# Psychophysiological and Neurobiological Responses to Deception and Emotional Stimuli: A Pilot Study on the Interplay of Personality Traits and Perceived Stress

**DOI:** 10.3390/brainsci15121252

**Published:** 2025-11-21

**Authors:** Andrei Teodor Bratu, Gabriela Carmen Calniceanu, Florin Zamfirache, Gabriela Narcisa Prundaru, Cristina Dumitru, Beatrice Mihaela Radu

**Affiliations:** 1Department of Anatomy, Animal Physiology and Biophysics, Faculty of Biology, University of Bucharest, Splaiul Independenţei 91-95, 050095 Bucharest, Romania; t.bratu20@s.bio.unibuc.ro (A.T.B.); carmen-gabriela.calniceanu@s.unibuc.ro (G.C.C.); zamfirache.florin@s.bio.unibuc.ro (F.Z.); prundaru.gabriela-narcisa@s.bio.unibuc.ro (G.N.P.); 2Department of Educational Sciences, Faculty of Educational Sciences, Social Sciences and Psychology, The National University of Science and Technology POLITEHNICA Bucharest, Pitești University Centre, Targul din Vale, 1, 110040 Pitesti, Romania

**Keywords:** deception, EEG, heart rate variability, personality traits, Dark Triad, perceived stress, frontal asymmetry

## Abstract

**Background/Objectives:** Deception engages both emotional and cognitive processes, yet individual variability in these responses remains insufficiently understood. This study aimed to investigate how personality traits, perceived stress, and empathic distress shape psychophysiological and neurobiological responses during deception and emotional processing. **Methods**: Thirty healthy young adults completed a protocol combining a deception task with emotional stimulus exposure, while heart rate (HR), heart rate variability (HRV), and electroencephalographic (EEG) activity were continuously recorded. Participants were characterized using measures of Dark Triad traits, perceived stress (PSS-10), and empathic distress. **Results:** The results showed increased HR and reduced HRV during deceptive responses, reflecting heightened cognitive effort and autonomic arousal. In contrast, morally or socially evaluative stimuli were associated with right-frontal EEG asymmetry, suggesting engagement of emotional regulation processes. Cluster analysis revealed distinct reactivity profiles: individuals with high stress and empathic distress exhibited amplified autonomic activation and reduced cortical inhibition, whereas those with higher Machiavellianism and psychopathy displayed attenuated HR/HRV modulation and stable EEG patterns, suggestive of emotional detachment and adaptive inhibition. These findings suggest that deception is a dynamic, context-dependent process influenced by individual personality traits and stress-regulation capacities. **Conclusions:** The study offers valuable insights into the psychophysiological mechanisms underlying deceptive behavior, with meaningful implications for both forensic and affective neuroscience.

## 1. Introduction

Deception is a multifaceted psychological process that relies on cognitive control, emotional regulation, and moral evaluation. These functions draw upon neurobiological systems also involved in stress and self-regulation, including the prefrontal cortex, anterior cingulate cortex, amygdala, and autonomic pathways [1,2,3]. Understanding deceptive behavior, therefore, requires examining how these systems operate under different emotional and personality contexts.

Stress plays a central role in shaping physiological and cognitive responses relevant to deception. Acute stress can enhance vigilance and adaptive responding, whereas chronic stress contributes to dysregulation, impaired emotion regulation, and weakened executive functioning [3,4]. When individuals engage in deceptive behavior, they often experience moral conflict and increased cognitive load, which manifests in psychophysiological changes such as elevated heart rate (HR), reduced heart rate variability (HRV), and alterations in frontal EEG asymmetry [5,6,7,8,9].

Emotional states further modulate these processes. Both negative and socially evaluative stimuli tend to increase sympathetic activation and right-frontal EEG activity. At the same time, positive emotions are more frequently associated with enhanced parasympathetic tone and left-frontal engagement [10,11]. When deception unfolds in emotionally charged or stressful contexts, these emotional and cognitive demands interact, producing complex patterns of autonomic and cortical activation.

Individual differences significantly influence how people respond to such situations. Personality traits associated with the Dark Triad—Machiavellianism, narcissism, and psychopathy—are linked to emotional detachment, strategic self-regulation, and lower physiological reactivity during morally or socially challenging tasks [12,13,14]. By contrast, individuals who experience high empathic distress or elevated perceived stress display stronger autonomic and cortical responses during deception, reflecting an intensified internal conflict [14,15,16,17].

Perceived stress adds another layer to this dynamic. Beyond external pressures, it captures subjective appraisal processes that influence autonomic flexibility and cortical activation patterns [15,16,17]. Higher levels of perceived stress have been associated with reduced HRV, increased sympathetic dominance, and shifts in frontal EEG asymmetry, particularly during tasks involving uncertainty or social evaluation.

In this context, the present pilot study examines how personality traits, perceived stress, and empathic distress interact to shape physiological and neural responses to deception and emotionally salient stimuli in young adults. By integrating HR, HRV, and EEG measures with validated psychological assessments, the study aims to identify individual reactivity patterns and provide preliminary insights into the cognitive–affective mechanisms underlying deceptive behavior.

## 2. Materials and Methods

### 2.1. Participants

Thirty healthy young adults (mean age = 24.6 years, SD = 3.1) participated voluntarily in the study. All participants had normal or corrected-to-normal vision and reported no history of neurological or psychiatric disorders, aligning with inclusion criteria commonly applied in pilot research on deception and stress-related psychophysiology [6,13,18]. Written informed consent was obtained prior to participation, and the protocol was approved by the Research Ethics Commission of the University of Bucharest (Decision no. 85/10.11.2022).

Given the modest sample size (*n* = 30) and gender imbalance (70% female), this study should be considered an exploratory pilot investigation. The findings offer preliminary insights into psychophysiological patterns related to deception and emotional processing and should not be generalized beyond the characteristics of the current sample.

### 2.2. Procedure

The experimental protocol consisted of two components: a structured deception task and an emotional-stimuli task, both administered in a quiet and controlled laboratory environment. Participants first completed the deception task, which was organized into six standardized question sets (Set 1–Set 6), each designed to elicit distinct cognitive, emotional, or self-regulatory responses. All participants received the six sets in a fixed sequence to ensure procedural consistency.

Set 1 consisted of neutral questions, including simple non-emotional prompts (e.g., daily routines, preferences) used to establish a physiological baseline. Set 2 included personal and autobiographical questions intended to evoke natural self-reflection without strong emotional or moral activation. Set 3 contained social-evaluative and moral-dilemma questions involving judgments or scenarios with potential social consequences, designed to induce evaluative processes and mild moral stress. Set 4 involved emotional-valence questions associated with positive, negative, or neutral affective themes, allowing comparisons of emotional reactivity. Set 5 consisted of personality-relevant questions reflecting tendencies aligned with Dark Triad traits (e.g., competitiveness, manipulation, self-interest), phrased indirectly to reduce response bias. Set 6 included deception questions in which participants were explicitly instructed to provide truthful or deceptive answers, enabling examination of physiological and neural markers of intentional deception.

After completing the deception task, participants proceeded to the emotional-stimuli task. In this phase, they viewed two sets of images presented for a controlled duration: one set of emotionally neutral images and one set of emotionally salient images.

Although conditions were counterbalanced across participants to mitigate order effects, this approach cannot fully eliminate potential contamination between the deception and neutral conditions. Specifically, completing a deception block first may heighten arousal, cognitive effort, or strategic monitoring, which could influence responses during a subsequent neutral block.

### 2.3. Psychophysiological Measures of Stress

Cardiac pulse (HR) and stress-related variability were assessed using the Garmin Fēnix^®^ 5 smartwatch (Garmin Ltd., Olathe, KS, USA), equipped with a wrist-mounted optical heart rate sensor (Garmin Elevate™, Olathe, KS, USA) based on photoplethysmography (PPG). The device continuously records cardiac signals at 1 Hz during both baseline and experimental phases. Stress levels derived from heart rate variability (HRV) were computed using Garmin’s proprietary stress algorithm, which evaluates short-term fluctuations of inter-beat intervals (IBI) to estimate autonomic nervous system activity. The stress index ranged from 1 (very low stress) to 100 (very high stress), providing an estimate of overall stress load. HRV is defined as the physiological variation in the time interval between consecutive heartbeats, expressed in milliseconds. During stress, inter-beat intervals shorten, whereas in relaxed states, they lengthen. Thus, HRV analysis provided valuable information on how the body adapts to external demands. Although standard HRV analysis is typically derived from electrocardiography (ECG) by detecting R-waves in the QRS complex and calculating successive R–R intervals, the Garmin Fēnix 5 uses PPG-derived IBIs to estimate HRV indices. Previous studies have reported moderate-to-strong correlations between Garmin’s stress algorithm and conventional HRV measures, particularly under resting conditions [19,20]. For the present study, HR and stress values were collected before and after each block of questions or image presentations. Participants were instructed to remain seated and minimize movement during recordings to ensure signal accuracy. All measurements were conducted in a controlled laboratory environment at a constant ambient temperature. Time-domain indices (mean HR, mean HRV) were computed according to established guidelines [21]. Standardized rest intervals were used to allow physiological values to return to a predefined baseline threshold, minimizing carryover effects and ensuring reliable comparisons across conditions.

### 2.4. EEG Recording and Analysis

Electroencephalographic (EEG) activity was collected with a Muse Headband 2 (256 Hz; InteraXon, Toronto, ON, Canada) at frontal and temporal sites. Data were filtered (0.5–50 Hz), segmented by task phase, and cleaned using automated artifact rejection for eye blinks and motion. Spectral power for alpha, beta, and theta bands was extracted using FFT, and frontal asymmetry indices (FAA and FBA) were computed to assess affective and cognitive control. The Muse system has been validated in prior studies, demonstrating its reliability for capturing both event-related EEG activity and resting-state brain signals in research settings [22,23].

### 2.5. Psychological Questionnaires

Participants completed the following questionnaires before the experimental tasks:

The Short Dark Triad (SD3) ([12]; Romanian adaptation by Dragoș Iliescu) was employed to assess three socially aversive traits—Machiavellianism, narcissism, and psychopathy—which are often studied as an overlapping constellation known as the Dark Triad.

The Responsive Distress Scale ([24] Romanian adaptation by Dragoș Iliescu) was used to evaluate participants’ tendency to experience empathic distress and emotional reactivity. Derived from the International Personality Item Pool (IPIP), this 10-item self-report scale assesses individual differences in emotional responsiveness to the suffering of others, a construct closely aligned with the Agreeableness and Emotionality domains of personality. Total scores were calculated by averaging item responses, with reverse scoring applied where appropriate. “Reactive distress” is considered a trait related to Emotional Intelligence (EI), specifically reflecting the tendency to experience and respond to negative emotions—particularly those of others—in ways that may be either adaptive or maladaptive.

The Perceived Stress Scale (PSS-10; [16]; Romanian adaptation by Popescu Marina, Văduva Cristina-Angela, and Gherman Alexandru) was used to assess subjective stress. The PSS-10 is a widely used psychological instrument designed to measure the degree to which individuals appraise their life situations as stressful. It consists of 10 self-report items evaluating how unpredictable, uncontrollable, and overwhelming respondents perceive their experiences over the last month. Participants rate each item on a 5-point Likert scale ranging from 0 (“Never”) to 4 (“Very often”). Scores were computed by reverse-coding four positively stated items (items 4, 5, 7, and 8) and summing all items, yielding a total stress score ranging from 0 to 40. Higher scores indicated greater levels of perceived stress. The PSS-10 has demonstrated good internal consistency, with Cronbach’s alpha values typically ranging between 0.78 and 0.91 across diverse populations.

All questionnaires were administered in Romanian, using validated translations. Internal consistency coefficients (Cronbach’s α) in the present sample exceed 0.70 for all scales.

To explore whether distinct psychological profiles could be identified within our sample, we conducted a two-step clustering procedure using principal component analysis (PCA) followed by K-means clustering. PCA was first applied to the standardized psychological measures (Dark Triad traits, perceived stress, and empathic distress) to reduce dimensionality and minimize redundancy across variables that are known to be intercorrelated [12,13,14]. The first two principal components captured the majority of variance and represented: (1) a Dark Triad–dominance axis, and (2) a stress–distress axis.

K-means clustering was then performed on the PCA component scores to identify naturally emerging participant groups without imposing arbitrary thresholds on raw scores. The optimal number of clusters (k = 3) was selected using silhouette coefficients and elbow (inertia) analysis. Cluster centroids were interpreted by examining their relative loading patterns across the two PCA components rather than raw scale scores, ensuring that cluster descriptions reflect objective, data-driven differences.

### 2.6. Data Analysis

Data was analyzed using JASP. Descriptive statistics were calculated for all variables. Repeated-measures ANOVAs were conducted to examine differences between deception and truth conditions, as well as between emotional valence conditions (positive, negative, neutral). Personality traits and perceived stress scores were included as covariates in mixed-model analyses. Pearson correlations were used to assess associations between questionnaire scores and physiological measures. Significance was set at *p* < 0.05, with Bonferroni corrections applied for multiple comparisons.

### 2.7. Research Hypotheses

Primary hypotheses:

**Hypothesis** **H1.**
*Emotional stimuli produce higher HR, lower HRV, and right-frontal EEG asymmetry compared to neutral stimuli.*


**Hypothesis** **H2.**
*Deceptive responses differ from truthful ones in HR, HRV, and EEG asymmetry.*


**Hypothesis** **H3.**
*Higher perceived stress and empathic distress predict stronger autonomic activation.*


**Hypothesis** **H4.**
*High Dark Triad traits predict reduced physiological reactivity and greater frontal control.*


## 3. Results

### 3.1. Psychological Profile of Participants

The psychological assessment aimed to characterize participants’ affective reactivity, perceived stress levels, and personality traits relevant to emotional and interpersonal regulation. Three main dimensions were examined: emotional responsiveness, perceived stress, and Dark Triad personality traits (Machiavellianism, Narcissism, Psychopathy), which together provide a comprehensive overview of the psychological background likely to influence physiological reactivity during subsequent experimental tasks. Due to the limited sample size (*n* = 30), all analyses should be interpreted with caution, as the reduced statistical power increases the likelihood that subtle effects may not have been detected. As such, the present findings are best regarded as preliminary.

The Responsive Distress Scale was used to evaluate individual differences in empathic and affective sensitivity. As shown in Figure 1, scores ranged from 1 to 10, with a pronounced peak around 5, indicating a moderate level of emotional reactivity across the sample. More than 60% of participants scored between 4 and 8, suggesting a predominance of medium-to-high empathic responsiveness. The KDE curve showed a slight rightward skew, pointing to a subgroup with elevated affective vulnerability. Given its sensitivity to interpersonal stress, this index provides a relevant basis for linking emotional reactivity with cardiovascular and EEG measures in subsequent analyses.

Gender-based differences were also observed. Figure 2 illustrates that female participants displayed higher median scores and greater dispersion on the Responsive Distress Scale compared to male participants, whose scores clustered around lower central values. This pattern aligns with previous findings indicating greater affective empathy and emotional responsiveness among women [25,26]. These differences may influence both the magnitude and variability of psychophysiological responses to emotionally salient stimuli.

The Perceived Stress Scale (PSS) further contextualized participants’ psychological states. As illustrated in Figure 3, scores ranged between 20 and 46, with most clustering in the 30–36 interval. Approximately two-thirds of participants scored above the cutoff of 28, indicating elevated levels of perceived stress. The distribution displayed a slight rightward skew, suggesting a subset of individuals with particularly high stress sensitivity. Gender comparisons (Figure 4) showed slightly higher median PSS values and broader variability in female participants, with a few high-value outliers suggesting individual stress vulnerability. Male participants exhibited a more compact and symmetric distribution. These findings are consistent with previous evidence that women report higher perceived stress, likely reflecting both psychosocial and cultural influences [16,27].

To examine personality dimensions, scores on the Short Dark Triad (SD3) were analyzed alongside distress and stress measures. The correlation matrix in Figure 5 shows clear positive associations among Machiavellianism, Narcissism, and Psychopathy. The strongest correlations were observed between Narcissism and Psychopathy (r = 0.56, *p* = 0.0011) and between Machiavellianism and Narcissism (r = 0.55, *p* = 0.0014), with a weaker but positive relationship between Machiavellianism and Psychopathy (r = 0.35). These results reinforce the theoretical view that Dark Triad traits share a common foundation rooted in manipulative strategies, egocentrism, and reduced empathic sensitivity [12,28].

In contrast, negative correlations were found between Responsive Distress and both Machiavellianism (r = –0.22, *p* = 0.24) and Narcissism (r = –0.24, *p* = 0.20), though these did not reach statistical significance (Table 1). Psychopathy showed no significant association with distress, suggesting potentially distinct mechanisms of affective inhibition. Perceived stress exhibited no meaningful correlations with any of the Dark Triad traits (r = 0.17, *p* = 0.38), indicating that situational or environmental factors may play a larger role in shaping everyday stress perception than stable personality dimensions.

Overall, these psychological measures reveal a participant group characterized by moderate empathic reactivity, elevated perceived stress levels, and a stable intercorrelation among Dark Triad traits. While affective empathy tends to be lower among individuals scoring higher on Machiavellianism and Narcissism, these personality features are not directly linked to perceived stress. This dissociation supports the notion that aversive personality traits and subjective stress perception reflect partially distinct psychological mechanisms [29]. These baseline profiles provide a meaningful framework for interpreting interindividual differences in physiological and neural responses to subsequent moral, social, and deceptive challenges.

### 3.2. Psychophysiological Reactivity Patterns: Cluster-Based Psychological Typologies and Cardiovascular Modulations

To better understand interindividual variability in autonomic reactivity, we first identified distinct psychological typologies within the sample and then examined their correspondence with heart rate (HR) and heart rate variability (HRV) dynamics across neutral and deceptive conditions. This integrative approach links personality and stress-related traits to physiological regulation, providing a foundation for interpreting differential responses to moral, affective, and deceptive contexts.

Figure 6 depicts the resulting three clusters projected in the PCA space. Cluster 0 (green) is characterized by low Dark Triad scores and moderate stress, suggesting a relatively balanced and adaptive emotional profile. Cluster 1 (orange) includes participants with elevated Dark Triad scores, particularly Machiavellianism and Narcissism, combined with moderate perceived stress—potentially indicating a strategic or defensive regulation style. Cluster 2 (blue) groups participants with high distress and perceived stress but low-to-moderate Dark Triad scores, suggesting heightened affective vulnerability and stress sensitivity. These clusters represent psychologically meaningful subgroups that are expected to differ in physiological reactivity during experimental stimulation.

Table 2 summarizes the demographic and psychological characteristics of these clusters. Cluster 0 contained the oldest participants (mean age = 28.7 years), with low Dark Triad scores and moderate stress levels. Cluster 1 included younger individuals (mean age = 23.7 years) with the highest Machiavellianism (29.1), Narcissism (28.3), and Psychopathy (21.5) scores, alongside lower distress and stress. Cluster 2 consisted of the youngest participants (mean age = 22.8 years), with the highest distress (7.67) and perceived stress (PSS = 36.9), and intermediate Dark Triad scores.

To examine physiological modulation as a function of both experimental condition and psychological cluster, HR and HRV responses were analyzed during neutral question exposure (Set 1) and deceptive responses (Set 6). HR and HRV are established indices of autonomic activation and regulatory dynamics [21,30,31].

Group-level analyses revealed that neutral questions elicited a significant HR increase, from 85.20 bpm at baseline to 90.23 bpm post-stimulus (ΔHR = +5.03 bpm, t = 3.617, *p* = 0.001), as shown in Table 3. HRV increased modestly (+3.97 ms), but this change did not reach significance (t = 1.388, *p* = 0.176), indicating heterogeneous individual responses. These results suggest that even emotionally neutral stimuli may trigger anticipatory or task-related autonomic activation, primarily reflected in HR.

In the deception condition, physiological patterns shifted. HR showed a slight decrease (−1.03 bpm), and HRV declined (−3.97 ms), though neither change reached statistical significance (Table 4). This absence of robust cardiovascular activation may reflect covert autonomic inhibition or compensatory regulation during deceptive responses [18].

Stratification by psychological cluster (Table 5) revealed clear differences in autonomic modulation. Cluster 0, with low Dark Triad scores, displayed the strongest HR increase under neutral conditions (+9.86 bpm) and a concurrent HRV decrease (−2.86 ms), indicating strong sympathetic activation. During deception, HR increased modestly (+1.29 bpm), while HRV declined further (−5.00 ms), suggesting discomfort or tension. Cluster 1, with high Dark Triad traits, exhibited a smaller HR increase under neutral conditions (+3.07 bpm) but a notable HRV rise (+5.50 ms), consistent with effective autonomic control or attenuated reactivity; during deception, both HR and HRV decreased. Cluster 2, characterized by high stress and distress, showed a moderate HR increase (+4.33 bpm) and the largest HRV increase (+6.89 ms) under neutral conditions, followed by a flattened response during deception, likely reflecting autonomic overload or regulatory fatigue.

Finally, correlations between physiological changes and psychological traits were examined (Table 6). A significant negative correlation emerged between psychopathy and ΔHR during neutral stimulation (r = −0.39, *p* = 0.0318), indicating that individuals with higher psychopathy scores exhibited reduced cardiac activation. Narcissism showed a similar but nonsignificant trend. HRV modulation during deception correlated negatively with Machiavellianism (r = −0.22, *p* = 0.241), suggesting reduced autonomic flexibility in highly manipulative individuals. A positive, albeit nonsignificant, association was observed between perceived stress and HRV reactivity (r = +0.20, *p* = 0.287), consistent with heightened autonomic responsivity in more stressed individuals.

Taken together, these results reveal distinct psychophysiological reactivity patterns linked to underlying psychological typologies. Individuals with lower Dark Triad scores show strong HR activation but less flexible HRV responses, consistent with direct sympathetic engagement. Those with high Dark Triad traits demonstrate attenuated HR reactivity and more controlled autonomic responses, possibly reflecting strategic emotional regulation. In contrast, participants with high stress and distress exhibit heightened reactivity under neutral conditions but blunted responses during deception, suggesting regulatory saturation. These patterns align with previous work on the relationship between aversive personality traits, stress perception, and autonomic regulation [32].

### 3.3. Correlations Between Personality Traits and Cardiovascular Modulations

Personality-linked autonomic regulation offers insight into how individuals manage physiological states under social and cognitive demands. The Dark Triad traits—Machiavellianism, Narcissism, and Psychopathy—are associated with emotional detachment and strategic interpersonal control [28,32]. These traits are theorized to buffer physiological reactivity in self-relevant or stressful contexts [18,33]. Here, we examine their influence on cardiovascular dynamics during neutral and deceptive conditions.

Under neutral conditions, all three Dark Triad traits were linked to reduced heart rate reactivity, with Machiavellianism showing a clear negative correlation with ΔHR (Figure 7a) and similar patterns emerging for Narcissism and Psychopathy (Figure 7b,c). This blunted cardiac response suggests diminished affective engagement or stronger top-down control, a pattern reinforced by cluster distributions: low-reactivity clusters exhibited minimal ΔHR regardless of trait scores, whereas higher-reactivity clusters showed greater cardiac acceleration. In contrast, heart rate variability displayed the opposite trend, with Machiavellianism positively correlated with ΔHRV (Figure 7d) and comparable, though weaker, associations for Narcissism and Psychopathy (Figure 7e,f). The divergence between reduced HR and preserved or increased HRV indicates that individuals with elevated Dark Triad traits may maintain effective parasympathetic regulation despite arousal cues, consistent with neurovisceral integration theory, which links higher HRV to more efficient executive and emotional control.

The analysis of autonomic responses during deceptive behavior was conducted to determine whether Dark Triad traits shape the physiological mechanisms involved in lying, offering insight into how personality influences emotional engagement, stress regulation, and strategic behavior. During deception, the relationships between these traits and autonomic activity became stronger and more distinct. Higher Machiavellianism was associated with decreased HR (Figure 8a), indicating suppressed sympathetic activation and reduced somatic stress signals, a pattern mirrored by Narcissism and Psychopathy (Figure 8b,c). These findings suggest that individuals high in Dark Triad traits experience lower emotional cost when deceiving and can inhibit physiological arousal to support smoother impression management [18,33]. HRV responses provided additional differentiation: Machiavellianism showed a positive correlation with ΔHRV (Figure 8d), reflecting preserved or enhanced autonomic flexibility that may facilitate controlled, strategic deception; Narcissism exhibited mixed HRV patterns (Figure 8e), likely influenced by self-presentational demands; whereas Psychopathy showed a negative association with ΔHRV (Figure 8f), indicating autonomic rigidity and reduced adaptive modulation. Overall, this analysis clarifies how distinct Dark Triad components shape the physiological architecture of deception, revealing whether lying is accompanied by strategic autonomic control, emotional disengagement, or low baseline arousal.

Taken together, these results outline a psychophysiological gradient across Dark Triad traits. Machiavellianism is characterized by blunted HR but preserved or enhanced HRV, reflecting strategic autonomic control and physiological composure. Narcissism is associated with dampened HR and heterogeneous HRV, indicating variable regulatory strategies depending on the situational context. Psychopathy, in contrast, is marked by blunted HR and reduced HRV, reflecting autonomic rigidity and low emotional engagement. These patterns reinforce the view that aversive personality traits modulate both the intensity and flexibility of cardiovascular responses, shaping how individuals react to affective, social, and deceptive demands. In deception contexts, this modulation likely facilitates reduced physiological detectability and increased behavioral fluency, offering an adaptive advantage in manipulative interactions.

### 3.4. Cardiovascular Reactivity to Moral, Social, and Personality-Provocative Stimuli

Understanding how different cognitive and emotionally salient stimuli influences autonomic responses provides important insight into the physiological mechanisms underlying moral reasoning, self-presentation, and perceived identity threat. In this part of the study, heart rate (HR) and HRV-based stress scores were recorded before and after each question set to assess cardiovascular changes associated with specific emotional and cognitive demands.

One of the central aims was to examine responses to moral dilemmas (Set 3), which included hypothetical scenarios involving honesty, altruism, or justice. Moral reasoning of this kind is known to engage prefrontal control systems and autonomic regulation through mechanisms of cognitive dissonance, moral conflict, and self-monitoring [10,30]. Physiological measurements revealed a modest HR increase from 87.93 bpm at baseline to 90.77 bpm after stimulation (Δ = +2.83 bpm, t = 1.576, *p* = 0.126), accompanied by a small, nonsignificant rise in the Garmin HRV Stress Score from 56.10 to 57.13 (Δ = +1.03; t = 0.427; *p* = 0.673) (Table 7). This pattern suggests mild sympathetic engagement without pronounced stress reactivity, consistent with a cognitive processing load rather than affective arousal. Such responses likely reflect substantial interindividual variability linked to empathic sensitivity, moral reasoning style, and stress vulnerability.

A contrasting physiological profile emerged for the social image set (Set 4), which targeted reputational concerns and self-presentation. Here, HR increased slightly from 86.77 bpm to 87.83 bpm (Δ = +1.07 bpm), while HRV stress showed a marginal decrease of 0.17 (from 56.43 to 56.27). These differences did not reach statistical significance (HR: t = 1.068, *p* = 0.294; HRV: t = −0.072, *p* = 0.943) (Table 8). The absence of measurable autonomic activation suggests that such socially evaluative content was processed in a controlled and familiar manner. This aligns with established models of habitual impression management, whereby individuals maintain physiological stability during self-presentation [25,32].

More pronounced autonomic responses were elicited by the Dark Triad set (Set 5), which focused on Machiavellianism, narcissism, and psychopathy—traits associated with socially aversive or identity-threatening content. Exposure to these items produced a significant HR increase from 86.40 bpm to 90.60 bpm (Δ = +4.20 bpm; t = 3.401, *p* = 0.002), whereas HRV stress remained stable (Δ = +0.43; t = 0.264; *p* = 0.793) (Table 9). This dissociation between HR and HRV mirrors a rapid, transient sympathetic activation in the absence of sustained parasympathetic withdrawal, a physiological pattern commonly observed during identity-threat processing and self-referential aversive content [30,34].

When comparing all five stimulus categories, clear differences emerged in both amplitude and significance of cardiovascular responses (Table 10). The largest HR increases were recorded during neutral questions (Set 1) (+5.03 bpm, *p* = 0.001), personal history (Set 2) (+4.83 bpm, *p* = 0.025), and Dark Triad traits (Set 5) (+4.20 bpm, *p* = 0.002), all of which were statistically significant. More modest and nonsignificant increases were observed during moral dilemmas (Set 3) (+2.83 bpm, *p* = 0.126) and social-image questions (Set 4) (+1.07 bpm, *p* = 0.294). By contrast, the HRV-derived stress index displayed only subtle, nonsignificant fluctuations across all conditions, with the largest change corresponding to neutral stimuli (+3.97; *p* = 0.176) and the largest decrease in personal history (−3.30; *p* = 0.103). These results indicate that HR is more sensitive to short-term autonomic shifts than HRV when measured with wearable sensors in subtle emotional-cognitive contexts.

Further numerical details are compiled in Table 11, which provides a comprehensive overview of ΔHR and ΔHRV values across all sets. The generally small and nonsignificant HRV variations underscore the stability of parasympathetic regulation under these task conditions, contrasting with the rapid cardiovascular responses captured by HR. This distinction highlights the complementary nature of the two measures: while HR reflects fast sympathetic activation, HRV is more suited for detecting prolonged or cumulative regulatory changes.

Figure 9 synthesizes these findings by depicting the relative magnitude of physiological changes across the five stimulus categories. HR responses were clearly highest in Sets 1, 2, and 5, reflecting anticipatory engagement, personal salience, or identity threat. In contrast, Sets 3 and 4 were associated with minimal physiological deviation from baseline, pointing to predominantly cognitive rather than affective processing. HRV remained largely stable across all conditions, confirming its insensitivity to transient sympathetic arousal.

Taken together, these cardiovascular data reveal a graded autonomic profile modulated by the emotional salience and self-relevance of the stimuli. Personally significant or identity-threatening content (Sets 2 and 5) elicited robust HR increases, moral dilemmas (Set 3) produced moderate but nonsignificant changes, and socially evaluative questions (Set 4) showed minimal impact on autonomic activity. Interestingly, neutral questions (Set 1) also induced strong HR activation, likely reflecting anticipatory arousal and task engagement, rather than affective salience per se. These patterns are consistent with psychophysiological evidence linking self-relevance and emotional depth to differential autonomic dynamics [10,25,30,32,34].

The combined interpretation of HR and HRV data provides a multilayered physiological perspective: HR functions as a sensitive index of rapid, stimulus-driven activation, while HRV captures more sustained parasympathetic modulation. This dissociation is particularly relevant for paradigms involving deception, moral reasoning, and self-evaluation, where subtle cognitive–affective factors shape autonomic reactivity. These cardiovascular findings establish an essential physiological baseline for subsequent integration with neural measures, such as frontal alpha asymmetry.

### 3.5. Frontal Hemispheric Activation Dynamics: Task-Dependent Variations in EEG Alpha/Beta Asymmetry

Frontal hemispheric lateralization plays a central role in the coordination between affective–motivational states, cognitive control, and physiological arousal. In this experimental phase, we examined frontal alpha (FAA) and frontal beta asymmetry (FBA) across six stimulus sets, integrating these cortical dynamics with autonomic indices—heart rate change (ΔHR) and HRV-derived stress variation (ΔStress). FAA was interpreted as an affective–motivational marker, reflecting relative left- or right-frontal engagement, while FBA was considered an index of approach-related cognitive activation and executive control.

A clear temporal structure emerged in the evolution of frontal asymmetry across task demands (Figure 10). In Sets 1 and 2, both FAA and FBA values were positive, indicating a predominantly left-frontal pattern associated with approach motivation, affective stability, and controlled engagement. Starting with Set 3 (moral dilemmas), FAA shifted below zero, signaling increasing right-frontal recruitment. This trend intensified through Sets 4 and 5—characterized by social-image challenges and Dark Triad provocations—before reaching its lowest values during Set 6 (forced deception). FBA followed a comparable trajectory: initial left dominance in Sets 1–2, a sharp inflection in Set 3, sustained negative deflections thereafter, and a partial—but not complete—rebound in Set 6. This progressive transition from left to right frontal dominance mirrors the mounting cognitive–affective load, moral ambiguity, and self-referential salience embedded in the experimental design.

The relationship between cortical and autonomic dynamics further illuminates these shifts (Figure 11). FAA correlated negatively with ΔHR across most sets, peaking at r = −0.47 in Set 4, indicating that greater right-frontal engagement coincided with stronger cardiac acceleration during socially evaluative conditions. FBA displayed negative correlations with ΔHR in Sets 1 and 2 (r = −0.41 and r = −0.36), consistent with efficient regulatory control under low-threat contexts. In Sets 4–5, however, this relationship reversed, becoming positive, suggesting approach-oriented mobilization under identity-relevant challenges. Correlations with ΔStress were generally weak, except in Set 6 where FAA showed a moderate negative correlation (r = −0.30), pointing to increased right-frontal dominance in the absence of a strong HRV-stress response—a profile consistent with physiological blunting during deception.

Cluster-level analyses provided additional resolution (Table 12). Cluster 0, marked by reduced HR (ΔHR_total = −9 bpm) and elevated stress (+15), reflected a cardiovascular inhibition profile coupled with heightened subjective load, possibly indicative of compensatory cortical control. Cluster 1 showed moderate increases in both HR (+2.88 bpm) and stress (+9.12), suggesting balanced activation, while Cluster 2 displayed the largest HR increase (+6.92 bpm) with the smallest stress increment (+6.08), a pattern consistent with high engagement and efficient autonomic buffering. These differentiated profiles underline the functional heterogeneity of cortical-autonomic coupling across participants.

A second correlation matrix (Figure 12) confirmed the robust FAA–ΔHR negative coupling across conditions, with a marked peak in Set 4. FBA–ΔHR correlations were negligible in neutral contexts but shifted weakly positive in response to provocative stimuli. FAA–ΔStress and FBA–ΔStress associations remained minimal overall, supporting the view that rapid cardiovascular adjustments are more closely aligned with FAA/FBA changes than slower HRV-derived stress indices.

Spatial mapping of participants in the FAA–FBA plane revealed three stable EEG clusters (Figure 13). Cluster 0 exhibited high beta activation and near-neutral alpha lateralization, reflecting strong cognitive drive with balanced affective tone. Cluster 1 occupied the upper-right quadrant (FAA+, FBA+), representing a left-frontal dominance pattern typically linked to approach motivation, regulatory efficiency, and adaptive engagement. Cluster 2, positioned in the negative domain of both FAA and FBA, indicated right-frontal predominance associated with withdrawal tendencies and stress vulnerability. These clusters remained stable across sets, suggesting trait-like EEG response profiles that interact with, but are not fully overridden by, task context.

Emotional valence further modulated these dynamics. Positive and neutral stimuli elicited a robust parasympathetic response, with significant reductions in HR (−4.50 bpm, *p* = 0.04) and ΔStress (−12.07, *p* = 0.001) (Table 13). In contrast, negative stimuli produced only a modest HR decrease (−2.27 bpm, *p* = 0.04) and a nonsignificant ΔStress change (+0.60, *p* = 0.77), consistent with orienting or inhibitory responses without significant stress escalation. These differences are clearly visualized in Figure 14, which illustrates the distinct autonomic signatures associated with positive versus negative affective content. The bar plot highlights the more pronounced vagal downregulation during positive/neutral conditions and the blunted HRV response to negative stimuli, reinforcing the view that valence-dependent autonomic modulation is asymmetric.

At a global level, cortico–autonomic coupling was modest (Table 14). FAA–ΔHR (r = +0.19) and FBA–ΔHR (r = +0.22) displayed weak positive trends, whereas FAA–ΔStress (r = −0.09) and FBA–ΔStress (r = −0.11) remained negligible. These small overall effect sizes conceal stronger, task-dependent relationships that emerged under specific affective and self-referential demands, reinforcing the importance of context-sensitive interpretations of frontal asymmetry and its physiological correlates.

Cluster-specific autonomic patterns further enriched this picture. During Set 1 (positive/neutral imagery), Cluster 1 displayed the most pronounced HR increase (+7.33 bpm) with a marked stress decrease (−24.67), reflecting high cortical engagement under efficient regulatory control (Table 15). Cluster 0 showed mild HR activation (+0.75 bpm) and moderate stress reduction (−8.25), whereas Cluster 2 exhibited a substantial HR decrease (−8.58 bpm) and moderate stress reduction (−11.68), indicative of vagal inhibition with low cortical drive.

Complementing these patterns, Table 16 presents correlations between frontal asymmetry and autonomic responses for Set 1. FAA was weakly and negatively associated with both ΔHR (r = −0.24) and ΔStress (r = −0.16), suggesting that greater right-frontal activity correlated with lower autonomic activation under non-threatening conditions. FBA showed virtually no relationship with ΔHR (r = +0.02) but a small positive correlation with ΔStress (r = +0.23), indicating subtle physiological mobilization linked to beta activity in positive contexts.

Under negative emotional stimulation (Set 2), the clustering structure diverged significantly (Figure 15; Table 17). Cluster 0 (FAA+/FBA+) maintained an adaptive regulation profile with moderate reductions in both HR and stress. Cluster 1 (FAA−/FBA−) exhibited minimal HR change and increased stress, consistent with right-frontal dominance and reduced autonomic flexibility, whereas Cluster 2 (FAA++/FBA+++)—although comprising a single individual—showed an extreme hyper-reactive pattern, combining marked cortical activation with heightened stress.

Cluster transitions between Sets 1 and 2 (Table 18) demonstrated dynamic EEG reorganization. Most participants from Cluster 0 redistributed between Clusters 0 and 1, suggesting two divergent regulatory pathways under affective load. The sole Cluster 1 participant from Set 1 transitioned to Cluster 2, displaying extreme activation. Participants from Cluster 2 moved to Cluster 0, indicating a shift toward more balanced frontal engagement. Figure 16 visualizes this redistribution, illustrating how task context dynamically reshapes frontal asymmetry without erasing individual EEG signatures.

In summary, these results reveal a highly structured interplay between frontal hemispheric asymmetry and autonomic regulation across escalating cognitive–affective task demands. During neutral or positively valenced conditions, left-frontal predominance aligns with low autonomic arousal and parasympathetic downregulation. As stimuli become morally complex, socially evaluative, or deceptive, right-frontal engagement intensifies, accompanied by increased HR but blunted HRV-derived stress modulation. Emotional valence exerts an additional layer of modulation, with positive content enhancing vagal regulation and negative content evoking mixed orienting and inhibitory responses.

## 4. Discussion

The present study investigated how personality traits—particularly those comprising the Dark Triad (Machiavellianism, narcissism, and psychopathy)—interact with perceived stress and empathic distress to modulate psychophysiological (HR, HRV) and neurophysiological (EEG) responses during deceptive behavior. Using a multimodal paradigm combining autonomic and cortical measures with validated psychological assessments, we captured context-dependent reactivity across neutral, emotional, moral, and self-referential tasks. This integrative design reflects the complex interplay between affective engagement, cognitive control, and dispositional factors, offering a more nuanced perspective on the physiological underpinnings of deception [5,35].

A central observation of this study provides clear support for Hypothesis 1, which predicted stronger autonomic and cortical activation during emotional compared to neutral contexts. Emotional stimulation elicited significant heart rate increases, subtle but consistent HRV modulations, and a shift toward right-frontal EEG asymmetry—an electrophysiological marker associated with withdrawal-related emotional processing. These findings are aligned with classical models of affective–autonomic coupling [10,30] and with more recent evidence on frontal asymmetry as a sensitive index of affective engagement [11], and with new empirical data highlighting the robustness of affective EEG markers in ecologically valid paradigms. The fact that even controlled laboratory stimuli produced measurable cardiovascular and cortical responses underscores the high sensitivity of these markers to the affective and moral salience of the context.

The results further support Hypothesis 2, according to which deception would elicit a distinctive physiological signature compared to truthful responding. Rather than an increase in physiological arousal, deceptive responses were characterized by a blunted HR and HRV profile, combined with stronger right-frontal EEG activation. This pattern is consistent with the recruitment of prefrontal control mechanisms involved in the strategic regulation and inhibition of stress-related autonomic signals during deceptive acts [6,18,36]. Moreover, recent neuroimaging evidence indicates that deception engages domain-general cognitive control and emotion regulation networks, allowing for differentiation from confounding states such as stress or conflict [37]. These data suggest that deception reflects an adaptive regulatory process rather than a simple hyperarousal response.

Regarding Hypothesis 3, which proposed that individuals with higher empathic distress and perceived stress would display greater autonomic reactivity, our findings offer only partial support. Participants with elevated distress levels exhibited greater cardiovascular lability, particularly in HRV, under both neutral and emotionally salient conditions. However, the magnitude and consistency of these effects varied considerably across individuals. This variability is in line with previous research showing that stress reactivity is modulated by coping style, appraisal processes, and regulatory [15,17], as well as recent evidence that trait vulnerability alone is an insufficient predictor of physiological stress reactivity without considering personality and temperament facets [38]. These findings suggest that trait vulnerability alone is insufficient to predict physiological reactivity without considering contextual and individual moderators.

Finally, Hypothesis 4 was supported by converging cardiovascular and EEG data. Individuals with higher Dark Triad scores exhibited attenuated HR responses and stable HRV patterns across experimental conditions, pointing to reduced autonomic activation. Psychopathy, in particular, showed a robust negative association with heart rate reactivity, consistent with literature linking this trait to emotional detachment and low physiological sensitivity to stressors [28,32,34]. Machiavellianism was associated with preserved or even enhanced HRV during deception, suggesting strategic parasympathetic regulation and emotional disengagement [12,39]. Recent neuroimaging findings further support these interpretations, showing that Dark Triad traits are associated with distinct neural signatures reflecting attenuated emotional engagement and increased regulatory control during socially relevant tasks [40].

Several limitations must be acknowledged when interpreting these results. First, a major limitation is that the relatively modest sample size may have reduced the power to detect more subtle interaction effects between traits, context, and physiological responses. While sample sizes of 20–40 participants are common in pilot psychophysiological and EEG research, they do not support definitive conclusions or broad generalizations. As such, the cluster analyses and trait–physiology associations should be interpreted as exploratory patterns rather than robust effects. Replication in substantially larger and more demographically balanced samples is necessary to validate these trends and enhance the reliability of the findings. Second, the use of wearable HRV sensors provided ecological validity but lower temporal precision compared to laboratory-grade ECG. Third, the structured deception paradigm cannot fully replicate the complexity and spontaneity of real-world lying. Finally, the cross-sectional nature of the study precludes causal inference regarding the directionality between personality traits and physiological reactivity. Future studies should address these limitations through larger and more heterogeneous samples, higher-resolution physiological measures, and more ecologically valid deception paradigms.

An additional limitation involves the potential contamination between the neutral and deception conditions. Although the presentation order was counterbalanced, exposure to a deception block may have influenced behavior or psychophysiological responses during a subsequent neutral block—potentially through sustained arousal, heightened vigilance, or increased cognitive control demands. While counterbalancing helps mitigate such carryover effects, it does not fully eliminate them. Future studies should consider using between-subjects designs, longer washout intervals, or independent condition presentations to more effectively prevent contamination.

A recommendation for future research is to replicate these findings using a between-subjects design for the deception manipulation [41], rather than a within-subjects counterbalanced approach. This would eliminate potential carryover effects between neutral and deceptive conditions, allowing for a clearer interpretation of condition-specific psychophysiological responses. Additionally, future studies should aim to recruit larger, more gender-balanced samples. A more demographically diverse cohort would increase statistical power, support more robust analyses of personality–stress interactions and improve the generalizability of the results.

Beyond sample-related limitations, repeated exposure to both deceptive and neutral conditions may have affected participant engagement, stress responses, or expectations. Such reactivity effects could either reduce the contrast between conditions or, alternatively, increase sensitivity to deceptive trials. Recognizing these dynamics is important for interpreting the current findings and underscores the need for future studies to adopt designs that minimize cross-condition contamination.

Another important methodological consideration involves the possibility that participants may infer the study’s purpose and adjust their behavior accordingly. Research on demand characteristics and participant awareness of experimental aims [41] indicates that such awareness can influence both behavioral and physiological responses. In deception paradigms, participants may become increasingly sensitive to what they believe is being evaluated, potentially amplifying, suppressing, or otherwise distorting the intended manipulation. To address this, future studies should incorporate explicit assessments of participant awareness—such as post-experimental questionnaires, funneled debriefings [42], or structured manipulation checks—to evaluate the extent to which participants detect the study’s purpose. Systematically examining these reactivity factors will help distinguish genuine deception-related effects from those shaped by participants’ interpretations of the research objectives.

Taken together, these findings advance current understanding of the dynamic interplay between emotional salience, deceptive intent, stress vulnerability, and personality traits in shaping physiological and neural responses. They also highlight that deception is not a uniform state characterized by generalized arousal, but a context-sensitive adaptive regulatory process modulated by stable personality dispositions. By integrating personality and stress markers with cardiovascular and EEG measures, this study provides a foundation for refining psychophysiological models of deception and for developing more precise, evidence-based approaches to the detection and interpretation of stress and deception in applied settings.

## 5. Conclusions

This study offered an integrative perspective on deception by combining physiological, neurophysiological, and psychological measures to explore how individuals respond to different types of questions. Deception elicits diverse physiological and neural responses shaped by individual personality and stress profiles. Autonomic and cortical data together reveal that lying is not purely a stress reaction but an adaptive process integrating emotional control and self-regulation. The most pronounced heart rate increases were observed during neutral and personal questions rather than explicit deception, suggesting that physiological activation may arise even in the absence of lying. EEG analyses revealed a shift toward right-frontal activation under morally or identity-threatening conditions, reflecting greater emotional engagement and regulatory demand. Personality traits, particularly psychopathy and Machiavellianism, modulated these responses—psychopathy being linked to reduced autonomic activation and Machiavellianism to increased physiological stress during loss-of-control situations. Although the cluster analysis suggested the presence of three participant groups with differing psychophysiological patterns, this analysis is exploratory in nature and should be interpreted with caution. Given the small sample size (N = 30), the resulting clusters do not represent stable or generalizable typologies but instead provide preliminary insights to guide future research.

## Figures and Tables

**Figure 1 brainsci-15-01252-f001:**
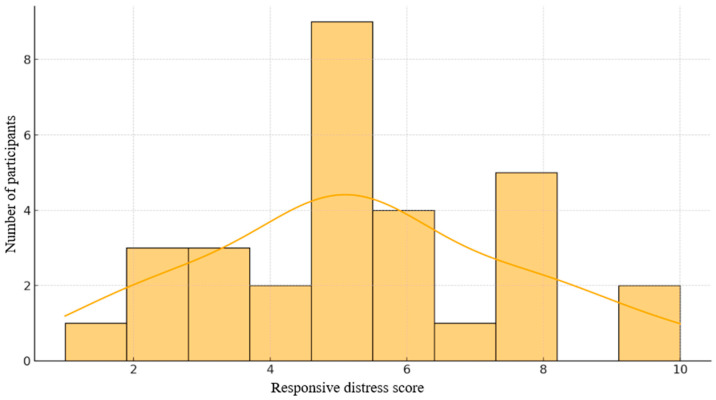
Distribution of scores on the Responsive Distress Scale.

**Figure 2 brainsci-15-01252-f002:**
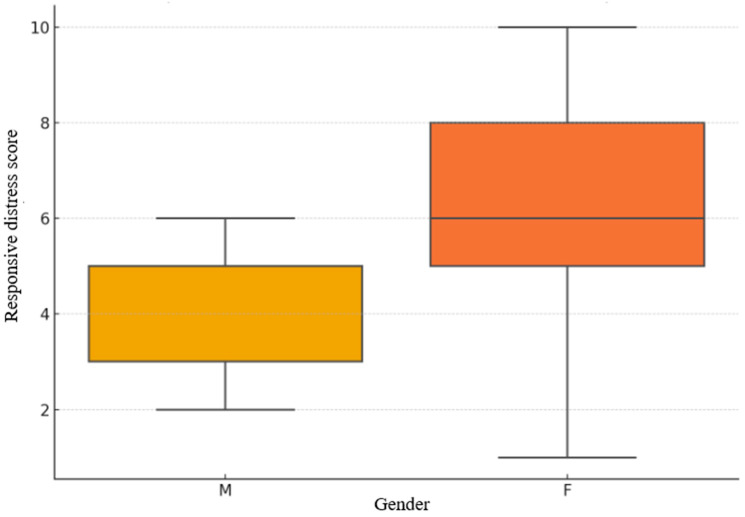
Gender-based distribution of Responsive Distress scores.

**Figure 3 brainsci-15-01252-f003:**
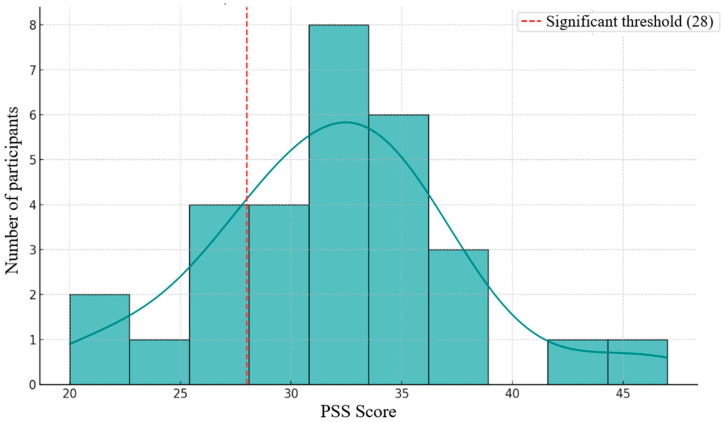
Distribution of scores on the Perceived Stress Scale (PSS).

**Figure 4 brainsci-15-01252-f004:**
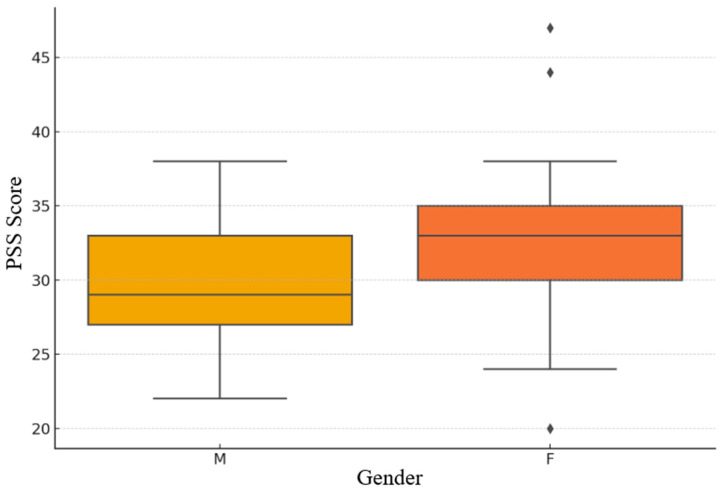
Gender-based distribution of PSS scores.

**Figure 5 brainsci-15-01252-f005:**
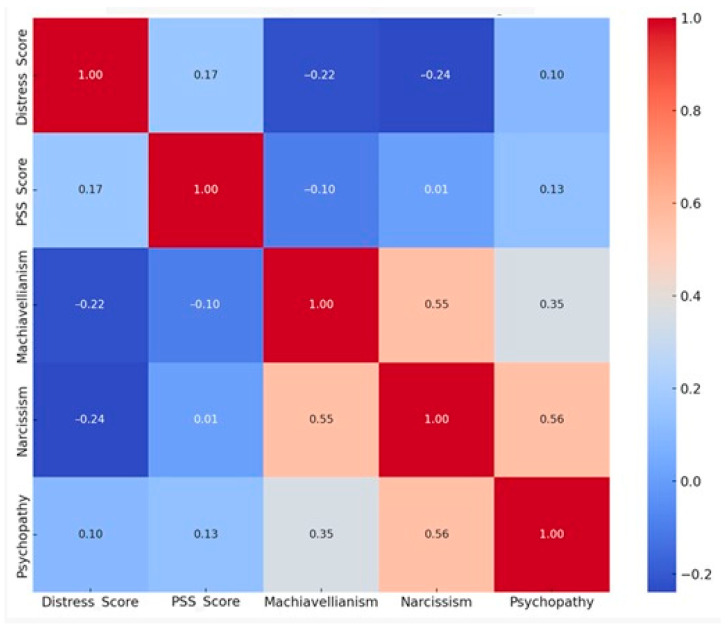
Correlation matrix among psychological scores.

**Figure 6 brainsci-15-01252-f006:**
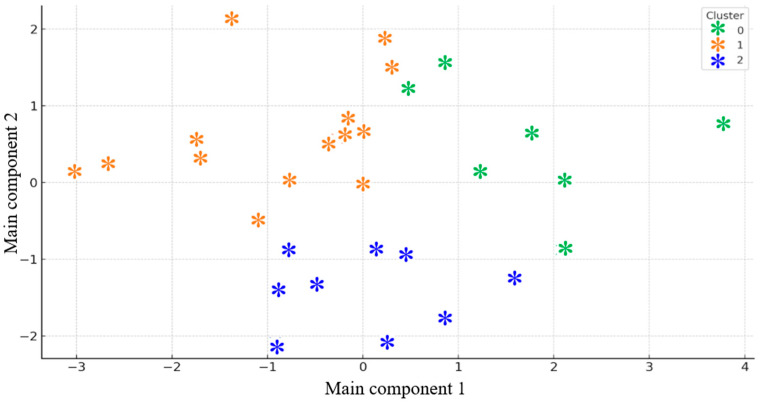
Clusters of participants based on psychological scores (PCA + K-means).

**Figure 7 brainsci-15-01252-f007:**
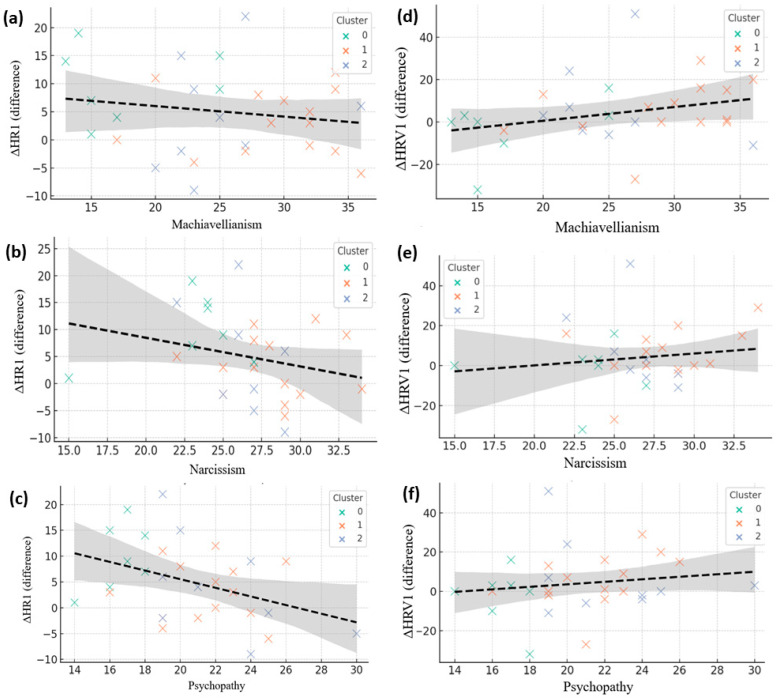
Correlations between Dark Triad traits and autonomic reactivity in neutral conditions. (**a**–**c**) Associations between Machiavellianism, Narcissism, and Psychopathy with heart rate change (ΔHR). (**d**–**f**) Corresponding associations with heart rate variability change (ΔHRV). Each point represents a participant, color-coded by psychological cluster (green = Cluster 0, orange = Cluster 1, blue = Cluster 2). Dashed lines indicate linear regression fits with shaded 95% confidence intervals.

**Figure 8 brainsci-15-01252-f008:**
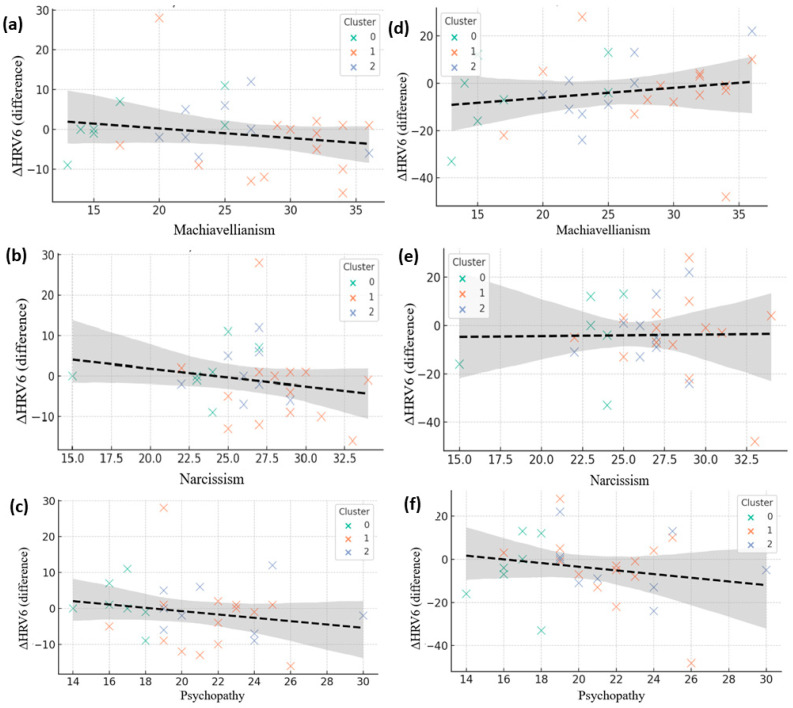
Correlations between Dark Triad traits and autonomic reactivity during deceptive conditions. (**a**–**c**) Associations between Machiavellianism, Narcissism, and Psychopathy with heart rate change (ΔHR). (**d**–**f**) Corresponding associations with heart rate variability change (ΔHRV). Each point represents a participant, color-coded by psychological cluster (green = Cluster 0, orange = Cluster 1, blue = Cluster 2). Dashed lines indicate linear regression fits with shaded 95% confidence intervals.

**Figure 9 brainsci-15-01252-f009:**
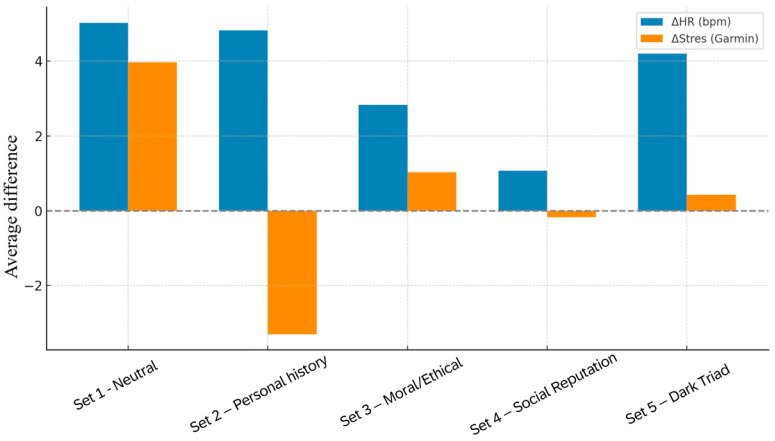
Comparative visualization of mean changes in heart rate (ΔHR, bpm) and HRV-derived stress (ΔStress) across all five question sets.

**Figure 10 brainsci-15-01252-f010:**
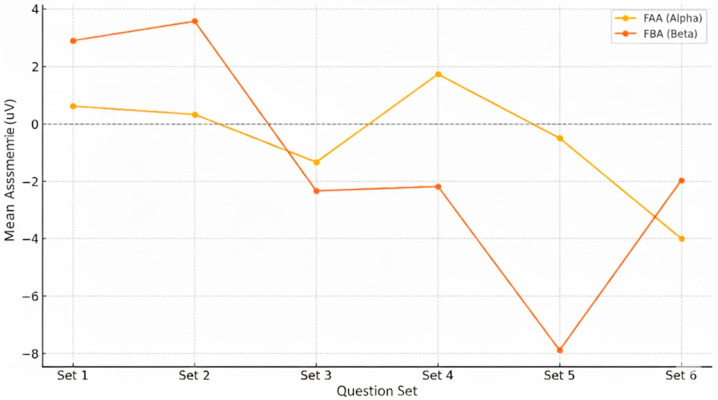
Evolution of frontal EEG asymmetry in Alpha (FAA) and Beta (FBA) across the six question sets. Values are mean right–left differences (μV) over frontal sites; positive = left dominance (reduced right activation), negative = right dominance (greater right activation).

**Figure 11 brainsci-15-01252-f011:**
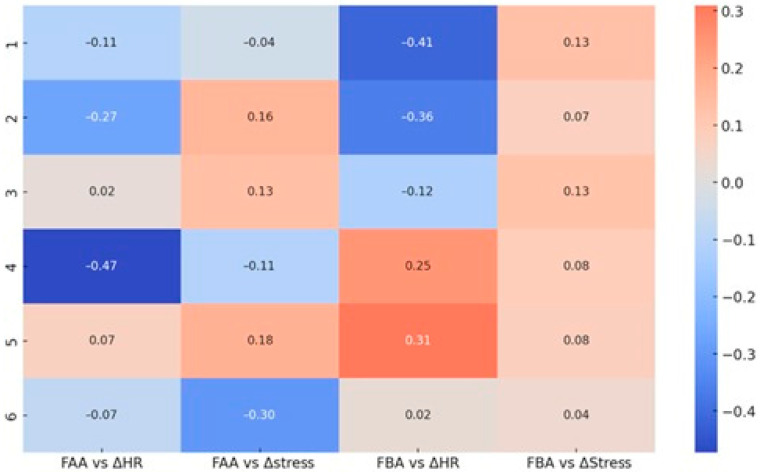
Correlation matrix linking frontal asymmetry (FAA—Alpha; FBA—Beta) with autonomic indices (ΔHR, ΔStress) for each question set. Cells display Pearson r; blue = negative, red = positive.

**Figure 12 brainsci-15-01252-f012:**
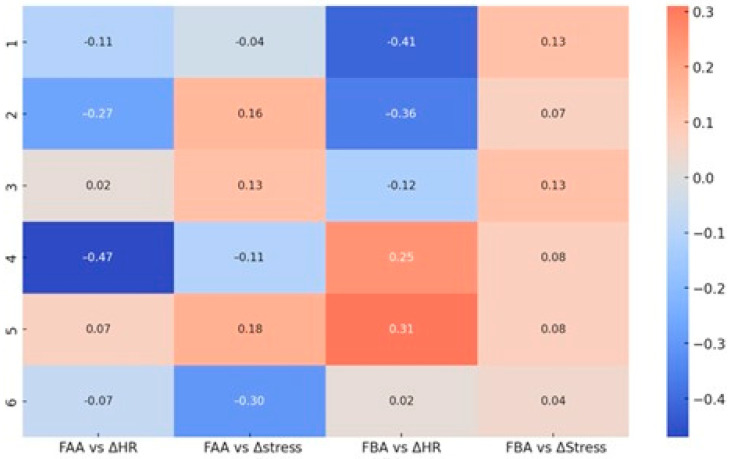
Correlation matrix between frontal EEG asymmetry (FAA—Alpha and FBA—Beta) and autonomic physiological parameters (ΔHR—heart rate, ΔStress—Garmin stress score) across the six question sets. Displayed values represent Pearson correlation coefficients, color-coded by the direction and magnitude of the relationship (red—positive, blue—negative).

**Figure 13 brainsci-15-01252-f013:**
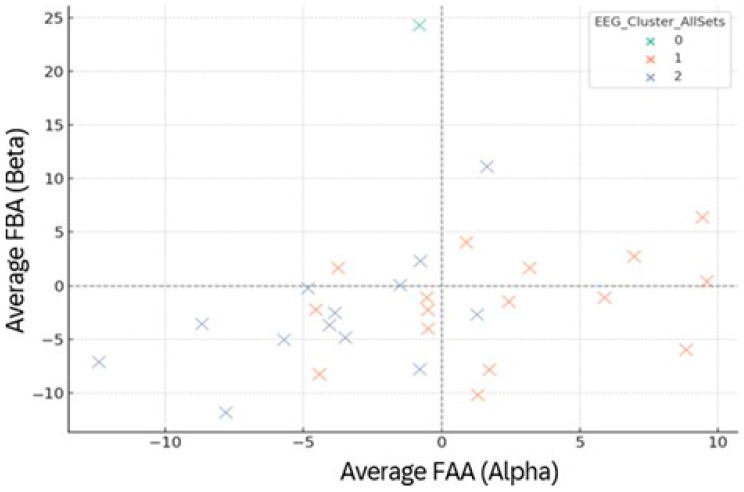
EEG clustering based on mean frontal asymmetry (FAA, FBA) across Sets 1–6. Points = participants, colored by EEG cluster (0/1/2). Dashed axes mark zero-dominance lines.

**Figure 14 brainsci-15-01252-f014:**
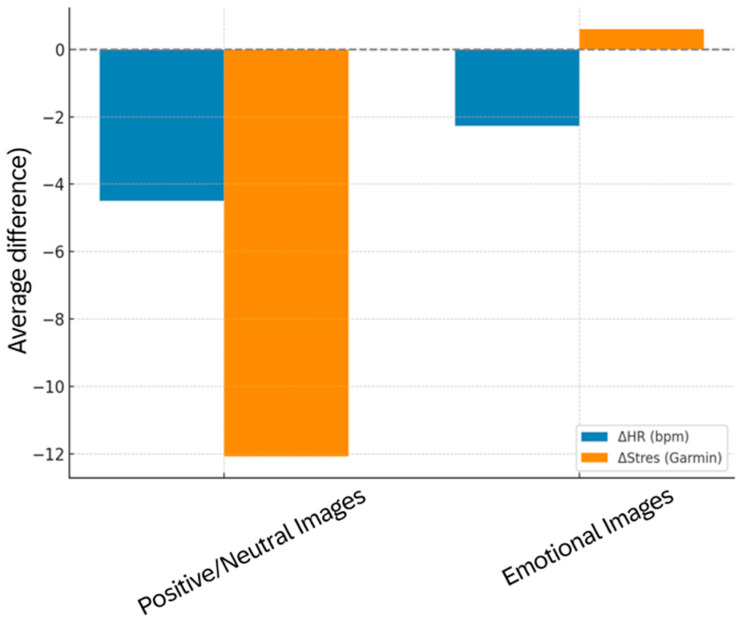
Bar plot of ΔHR (bpm) and ΔStress during exposure to positive/neutral vs. negative images. Negative values indicate reductions from baseline.

**Figure 15 brainsci-15-01252-f015:**
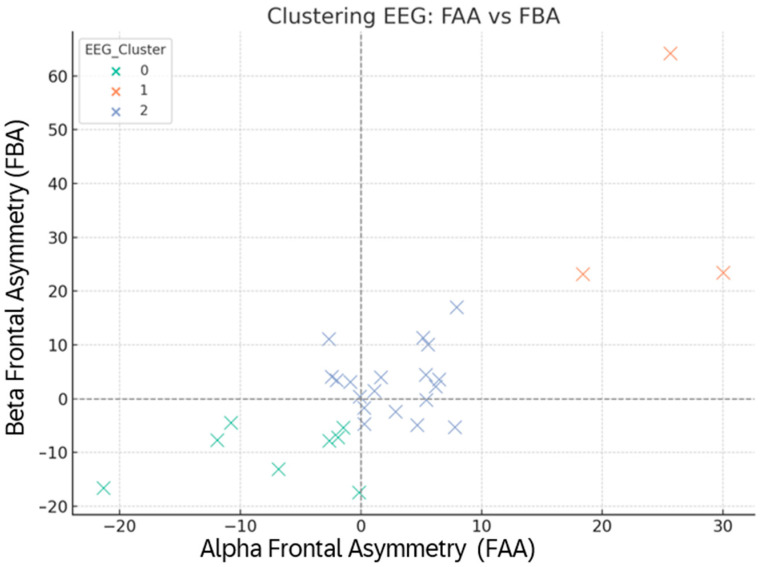
FAA–FBA clustering during negative images (Set 2). Colors: 0 = green, 1 = orange, 2 = blue. Dashed lines mark zero points on each axis.

**Figure 16 brainsci-15-01252-f016:**
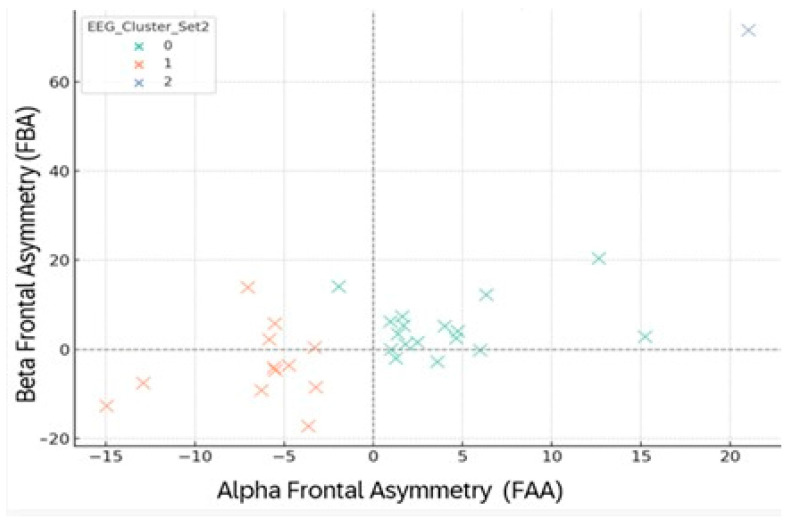
Two-dimensional representation of EEG clustering based on frontal Alpha (FAA) and Beta (FBA) asymmetry during the viewing of emotional images (Set 2). Each point represents a participant, color-coded according to EEG cluster membership (0—green, 1—orange, 2—blue). The axes indicate the mean FAA and FBA values, expressed in microvolts (μV). The dotted lines mark the zero reference thresholds, delineating the balance point between left and right hemispheric activation.

**Table 1 brainsci-15-01252-t001:** Pearson correlation coefficients (r) and *p*-values for psychological measures.

Correlation	R	*p*
Distress ↔ PSS	0.17	0.382
Distress ↔ Machiavellianism	–0.22	0.240
Distress ↔ Narcissism	–0.24	0.201
Narcissism ↔ Psychopathy	0.56	0.001
Machiavellianism ↔ Narcissism	0.55	0.001

**Table 2 brainsci-15-01252-t002:** Average demographic and psychological characteristics of participants by cluster (K-means analysis).

Cluster	Age	Mean	Distress	PSS	Machiavellianism	Narcissism Psychopathy
0	28.7	4.86	31.9	17.7	23.0	16.6
1	23.7	4.07	29.1	29.1	28.3	21.5
2	22.8	7.67	36.9	25.0	26.4	22.3

**Table 3 brainsci-15-01252-t003:** Mean changes in HR and HRV following presentation of neutral questions.

Parameter	Baseline Mean	Post-Stimulus Mean	Mean Change	t	*p*
HR (bpm)	85.20	90.23	+5.03 bpm	3.617	0.001
HRV (ms)	51.57	55.53	+3.97 ms	1.388	0.176

**Table 4 brainsci-15-01252-t004:** Mean changes in HR and HRV following presentation of deceptive responses.

Parameter	Baseline Mean	Post-Stimulus Mean	Mean Change	t	*p*
HR (bpm)	89.03	88.00	−1.03 bpm	−0.652	0.519
HRV (ms)	58.23	54.27	−3.97 ms	−1.407	0.170

**Table 5 brainsci-15-01252-t005:** Mean changes in HR (ΔHR) and HRV (ΔHRV) across experimental conditions neutral questions (set 1) and deceptive responses (set 6), stratified by psychological clusters.

Cluster	ΔHR1	ΔHRV1	ΔHR6	ΔHRV6
0	+9.86	−2.86	+1.29	−5.00
1	+3.07	+5.50	−2.64	−4.14
2	+4.33	+6.89	−0.33	−2.89

**Table 6 brainsci-15-01252-t006:** Pearson correlations (r) between physiological changes (ΔHR, ΔHRV) and psychological traits across experimental conditions (1 and 6).

ΔParameter	Trait	R	*p*
ΔHR1	Psychopathy	−0.39	0.0318
ΔHR1	Narcissism	−0.25	0.176
ΔHRV6	PSS	+0.20	0.287
ΔHRV6	Machiavellianism	−0.22	0.241

**Table 7 brainsci-15-01252-t007:** Mean changes in heart rate (HR) and HRV-derived stress score (Garmin HRV) following exposure to moral and ethical dilemma questions (Set 3).

Parameter	Baseline Mean	Post-Stimulus Mean	Mean Change	t	*p*
HR (bpm)	87.93	90.77	+2.83	1.576	0.126
HRV Stress Score	56.10	57.13	+1.03	0.427	0.673

**Table 8 brainsci-15-01252-t008:** Mean changes in HR and HRV-derived stress score (Garmin HRV) following exposure to socially sensitive questions.

Parameter	Baseline Mean	Post-Stimulus Mean	Mean Change	t	*p*
HR (bpm)	86.77	87.83	+1.07	1.068	0.294
HRV Stress Score	56.43	56.27	−0.17	−0.072	0.943

**Table 9 brainsci-15-01252-t009:** Mean changes in HR and HRV-derived stress score (Garmin HRV) following exposure to questions on Dark Triad traits.

Parameter	Baseline Mean	Post-Stimulus Mean	Mean Change	t	*p*
HR (bpm)	86.40	90.60	+4.20	3.401	.002
HRV Stress Score	54.27	54.70	+0.43	0.264	.793

**Table 10 brainsci-15-01252-t010:** Comparative summary of mean HR and HRV-derived stress changes (Garmin HRV Stress Score) across the five stimulus sets.

Set	ΔHR (bpm)	*p* (HR)	ΔStress (Garmin)	*p* (Stress)
Neutral Questions	+5.03	0.001	+3.97	0.176
Personal History	+4.83	0.025	−3.30	0.103
Moral/Ethical Dilemmas	+2.83	0.126	+1.03	0.673
Social Image	+1.07	0.294	−0.17	0.943
Dark Triad Traits	+4.20	0.002	+0.43	0.793

**Table 11 brainsci-15-01252-t011:** Mean changes in heart rate (ΔHR, bpm) and stress index (ΔStress, Garmin) across the five question sets, along with their corresponding *p*-values.

Set	ΔHR (bpm)	*p* (HR)	ΔStress (Garmin)	*p* (Stress)
Set 1—Neutral Questions	+5.03	0.001	+3.97	0.176
Set 2—Personal History	+4.83	0.025	−3.30	0.103
Set 3—Moral/Ethical Dilemmas	+2.83	0.126	+1.03	0.673
Set 4—Social Image	+1.07	0.294	−0.17	0.943
Set 5—Dark Triad Traits	+4.20	0.002	+0.43	0.793

**Table 12 brainsci-15-01252-t012:** Aggregated physiological profile (ΔHR_total, ΔStress_total) for EEG clusters across all sets.

EEG_Cluster_AllSets	ΔHR_Total (bpm)	ΔStress_Total
0	−9.00	+15.00
1	+2.88	+9.12
2	+6.92	+6.08

**Table 13 brainsci-15-01252-t013:** Mean autonomic changes while viewing positive/neutral vs. negative emotional images.

Set	ΔHR (bpm)	*p* (HR)	ΔStress	*p* (Stress)
Positive/Neutral	−4.50	0.04	−12.07	0.001
Emotional-Negative	−2.27	0.04	+0.60	0.770

**Table 14 brainsci-15-01252-t014:** Pearson correlations between frontal asymmetry (FAA, FBA) and autonomic indices (ΔHR, ΔStress), pooled across sets.

Correlation	R
FAA vs. ΔHR	+0.19
FAA vs. Δstress	−0.09
FBA vs. ΔHR	+0.22
FBA vs. Δstress	−0.11

**Table 15 brainsci-15-01252-t015:** Mean frontal asymmetry (μV) and autonomic change during positive/neutral images (Set 1).

EEG Cluster	FAA (α, μV)	FBA (β, μV)	ΔHR (bpm)	ΔStress
Cluster 0	−7.12	−9.99	+0.75	−8.25
Cluster 1	+24.70	+36.89	+7.33	−24.67
Cluster 2	+2.78	+2.97	−8.58	−11.68

**Table 16 brainsci-15-01252-t016:** Correlations between frontal asymmetry (FAA, FBA) and autonomic change (ΔHR, ΔStress) during positive/neutral images (Set 1).

Correlation	R
FAA vs. ΔHR	−0.24
FAA vs. Δstress	−0.16
FBA vs. ΔHR	+0.02
FBA vs. Δstress	+0.23

**Table 17 brainsci-15-01252-t017:** Mean EEG asymmetry and autonomic change by EEG cluster during negative images (Set 2).

EEG Cluster	FAA (α, μV)	FBA (β, μV)	ΔHR (bpm)	ΔStress
Cluster 0	+3.97	+4.78	−3.35	−3.53
Cluster 1	−6.54	−3.78	−0.50	+4.67
Cluster 2	+21.02	+71.56	−5.00	+22.00

**Table 18 brainsci-15-01252-t018:** Transitions between EEG clusters from Set 1 (neutral) to Set 2 (emotional).

Set-1 Cluster	Set-2 Cluster 0	Set-2 Cluster 1	Set-2 Cluster 2
Cluster 0	14	12	0
Cluster 1	0	0	1
Cluster 2	3	0	0

## Data Availability

The original contributions presented in this study are included in the article. Further inquiries can be directed to the corresponding author.

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
