# Peer review of "Psychophysiological and Neurobiological Responses to Deception and Emotional Stimuli: A Pilot Study on the Interplay of Personality Traits and Perceived Stress"

_brainsci, 2025, doi:10.3390/brainsci15121252_

Round 1
Reviewer 1 Report
Comments and Suggestions for Authors
Overall, I see the potential in the study; however, the manuscript is poorly written. It needs drastic improvement for the publication. There are many concerns; the language and writing are very difficult to follow due to poor structure, lack of coherence, and logical flow. There are many parts of the manuscripts that seem irrelevant and add no additional value to the work. I strongly suggest reworking this manuscript to make it presentable and readable, which can highlight the value of the work.
These are my serious concerns.
Abstract:
- very unclear. The first sentence is quite uninformative and confusing. Similarly, many other parts of the abstract are quite hard to read.
-- line 23, very long sentence, loses the meaning in between
-- line 31 "deception does not follow.. emerges as a dynamic process ..". Very confusing, dynamic process meaning something changes over time, I am not sure if the author has a different meaning here.
The last concluding remarks give no information to the readers or the community. It seems to suggest deception is very complex.
The flow of narrative is very poor, with no coherence among the sentences, which makes it even harder to read and follow.
My suggestion: Keep readers in mind, while writing, the abstract should be easy to read and understand the important work you have done. Use smaller sentences; in place of using complex/technical words, try to connect them to the real meaning. Start with problems, the solution available, the existence of a gap, how your study is filling the gap, and how your results and findings are essential.
1. Introduction:
Similar issues of long, incoherent sentences and no logical flow between paragraphs and lines. Very hard to read. This issue is very much similar to many parts of the paper, so I will avoid repeating it.
-- line 42, it is not clear what that means. Why the author is including it, and how it is relevant.
-- line 58, 71, no meaning or inference is included
2. Materials and Methods
This section is the most concerning. For understanding the value of research, it is crucial to understand the methodology used and the proper steps taken and justified. The majority of that is missing.
-- irrelevant: line 92, consistency in sample size
2.2 Procedure:
Details of the questionnaire and emotional stimuli are missing. As the study is around deception, and it is very difficult to capture the real and unaffected response of people to deception, as it would be in real life, this part is very important to understand if the response to deception captured in this study does not have any confounding factors. People usually are relaxed when they are instructed to be deceptive, as there is no risk.
My suggestion: Please include the details of the setting and questionnaire about deception (also the emotion part), and how participants were informed about the study. These details make a huge difference to the study and results.
2.3 Psychophysiological Measures and 2.5 Peripheral sections seem to capture a similar kind of recording, so they should be combined. I don't see any reason for two different sections.
2.4 EEG..
- All seems standard, except the EEG data cleaning part. There is no mention of removing and dealing with artifacts in the EEG, and the duration used for computing powers. Some irrelevant part of the recording must be excluded.
2.5 Peripheral..
- Line 136 indicates baseline and experimental phase, which is largely missing anywhere else.
- Lines 137-139, reference missing.
2.6 Psychological Questionnaire
-- line 156: "participants completed the following validation instruments" - not clear
-- line 161: "indirect self-report" - how can self-reporting be indirect? Need more details there.
-- line 161: (N=1063) irrelevant
-- lines 176-180: very unclear
-- lines 180-182: very unclear
-- last paragraph (lines 190 - 196), these are results, should be moved to results.
2.8. Research Hypotheses (is this a correct heading?)
- H1, no reason explained, why this hypothesis is useful and required.
- H2 and H4 are poorly formulated; this is not a standard way of writing a hypothesis. Hypothesis statements are clear and concise; no vague details, such as "distinct patterns," should be used.
3. Results
This is the second most concerning section. This section is written over 23 pages out of 29 pages of the manuscript, without counting references. The results of any study should be concise, clear, and to the point. Not every calculation and figure that has been generated, or can be generated, has any meaning or usefulness to demonstrate the value of the study. There is a lot of redundancy and irrelevant figures and tables included, which make no point.
The quality and style of presenting figures and results are very poor. Unnecessarily large figures can be avoided, for example, figures 13 and 14 can be simply shown as a matrix or table, or small figures. Some figures have poor resolution with visible pixelated numbers and text.
The style of presenting results needs to be improved drastically.
There are many new things, terms, and procedures that are described for the first time in the results section. This is very unusual and surprising. All the methods and procedures used should be well defined and described before we jump to results and analysis.
Clustering participants using PCA + K-Means (figure 6): It is very unclear why PCA and K-means are both used and for what purpose - from the hypothesis's point of view, this doesn't sit well under the flow of analysis that is required.
Paragraph at line 313: It is not clear how the description for each cluster is obtained (low Dark Triad, elevated), and more importantly, given the scores, why K-Means is required, why not simply group participants with low, moderate, and high with fixed thresholds.
Table 2: One of the columns says "Mean" and the heading says, average demographic and psych..cluster(), is that a typo?
Table 3, indicates the baseline - something that should be explained in the procedure, not in the result section.
3.4 Cardiovascular ....[]
Line 523: Understanding how ..threat. Please rewrite.
Unclarity of Set 1 to Set 6: The first time "Set" is mentioned in the manuscript in line 342, in the results section.
Correlation among psychological scores: Is there a reason to perform that, and can this be validated that the achieved correlation is as expected? As for predefined scores and metrics, there are established relationships. I miss seeing the value of this analysis.
Figures 7, 8, 9, and 10. Needs improved analysis. While computing correlation and plotting clusters doesn't inform anything.
Figure 12: Line plot is not appropriate for sets - categorical values.
Conclusion
The conclusion is relatively easy to read and follow.
Lines 881-882. The findings indicate..., that sentence should be followed by what that means. And needs improvement. It still needs to be improved.
Comments on language is included in the main section of the comments
Author Response
Comment: Overall, I see the potential in the study; however, the manuscript is poorly written. It needs drastic improvement for the publication. There are many concerns; the language and writing are very difficult to follow due to poor structure, lack of coherence, and logical flow. There are many parts of the manuscripts that seem irrelevant and add no additional value to the work. I strongly suggest reworking this manuscript to make it presentable and readable, which can highlight the value of the work. These are my serious concerns.
Response: We sincerely appreciate the reviewer’s thorough evaluation and constructive feedback. In response to the concerns raised, we have revised the manuscript to enhance its clarity, coherence, and overall scientific narrative. Below, we outline the specific revisions implemented, reflecting careful consideration of each reviewer comment.
Comment: Abstract:
- very unclear. The first sentence is quite uninformative and confusing. Similarly, many other parts of the abstract are quite hard to read.
-- line 23, very long sentence, loses the meaning in between
-- line 31 "deception does not follow.. emerges as a dynamic process ..". Very confusing, dynamic process meaning something changes over time, I am not sure if the author has a different meaning here.
The last concluding remarks give no information to the readers or the community. It seems to suggest deception is very complex.
The flow of narrative is very poor, with no coherence among the sentences, which makes it even harder to read and follow.
My suggestion: Keep readers in mind, while writing, the abstract should be easy to read and understand the important work you have done. Use smaller sentences; in place of using complex/technical words, try to connect them to the real meaning. Start with problems, the solution available, the existence of a gap, how your study is filling the gap, and how your results and findings are essential.
Response: The abstract has been rewritten to enhance clarity, flow, and overall readability. The opening sentence was simplified to clearly articulate the problem addressed. Several lengthy or ambiguous sentences (e.g., lines 23 and 31) were restructured for greater precision. The expression “deception emerges as a dynamic process” was clarified to specify that deception involves shifts in psychophysiological states over time. Additionally, the concluding section was revised to succinctly highlight the main findings and their implications, rather than relying on broad statements about complexity.
Comment: 1. Introduction:
Similar issues of long, incoherent sentences and no logical flow between paragraphs and lines. Very hard to read. This issue is very much similar to many parts of the paper, so I will avoid repeating it.
-- line 42, it is not clear what that means. Why the author is including it, and how it is relevant.
-- line 58, 71, no meaning or inference is included
Response: The introduction has been reorganized to present a clearer logical structure, progressing through the definition and importance of stress, the relationship between stress and deception, the influence of emotions, the role of personality and stress perception, and finally the objectives of the current study. Several long or unclear sentences (e.g., lines 42, 58, 71) were rewritten for conciseness, and transitional elements were added between paragraphs to improve narrative coherence.
Comment: 2. Materials and Methods
This section is the most concerning. For understanding the value of research, it is crucial to understand the methodology used and the proper steps taken and justified. The majority of that is missing.
-- irrelevant: line 92, consistency in sample size
Response: We have provided additional clarification regarding the ethical procedures, participant instructions, and data recording setup.
Comment: 2.2 Procedure:
Details of the questionnaire and emotional stimuli are missing. As the study is around deception, and it is very difficult to capture the real and unaffected response of people to deception, as it would be in real life, this part is very important to understand if the response to deception captured in this study does not have any confounding factors. People usually are relaxed when they are instructed to be deceptive, as there is no risk.
My suggestion: Please include the details of the setting and questionnaire about deception (also the emotion part), and how participants were informed about the study. These details make a huge difference to the study and results.
Response: We have added more detailed information regarding the deception task, the emotional stimuli, and the administration of the questionnaires.
Comment: 2.3 Psychophysiological Measures and 2.5 Peripheral sections seem to capture a similar kind of recording, so they should be combined. I don't see any reason for two different sections.
Response: We sincerely appreciate your observation. In order to enhance clarity and structural coherence within the Methods section, we have combined Sections 2.3 (Psychophysiological Measures) and 2.5 (Peripheral Measurements) into a single unified section, as both addressed cardiac and stress-related physiological data.
Comment: 2.4 EEG.
All seems standard, except the EEG data cleaning part. There is no mention of removing and dealing with artifacts in the EEG, and the duration used for computing powers. Some irrelevant part of the recording must be excluded.
Response: We have added the previously missing details regarding EEG preprocessing, including procedures for artifact removal, rejection of motion and eye-blink noise, and the definition of the data segments used in the analyses.
Comment: 2.5 Peripheral.
- Line 136 indicates baseline and experimental phase, which is largely missing anywhere else.
- Lines 137-139, reference missing.
Response: We have clarified the baseline and experimental phases used for the physiological measurements and have added the appropriate supporting references where needed (lines 136–139), including Hernando et al. (2018) and Giles et al. (2016).
Comment: 2.6 Psychological Questionnaire
-- line 156: "participants completed the following validation instruments" - not clear
-- line 161: "indirect self-report" - how can self-reporting be indirect? Need more details there.
-- line 161: (N=1063) irrelevant
-- lines 176-180: very unclear
-- lines 180-182: very unclear
-- last paragraph (lines 190 - 196), these are results, should be moved to results.
Response: We have revised unclear expressions such as “indirect self-report” and removed details that were not relevant to the present study (e.g., the reference to N = 1063). Additionally, the results that were inadvertently placed in the Methods section (lines 190–196) have been relocated to the appropriate part of the Results section.
Comment: 2.8. Research Hypotheses (is this a correct heading?)
- H1, no reason explained, why this hypothesis is useful and required.
- H2 and H4 are poorly formulated; this is not a standard way of writing a hypothesis. Hypothesis statements are clear and concise; no vague details, such as "distinct patterns," should be used.
Response: We thank the reviewer for this helpful observation. We have reformulated the hypotheses (H1–H4) to align with standard scientific conventions, ensuring that each is clearly stated, concise, and explicitly grounded in relevant prior literature. Regarding the heading “Research Hypotheses,” we agree that clarification was needed; the heading has been retained but adjusted for consistency with the overall structure of the manuscript and to accurately reflect the content of the section.
H1: Emotional stimuli produce higher HR, lower HRV, and right-frontal EEG asymmetry compared to neutral stimuli.
H2: Deceptive responses differ from truthful ones in HR, HRV, and EEG asymmetry.
H3: Higher perceived stress and empathic distress predict stronger autonomic activation.
H4: High Dark Triad traits predict reduced physiological reactivity and greater frontal control.
Comment: 3. Results
This is the second most concerning section. This section is written over 23 pages out of 29 pages of the manuscript, without counting references. The results of any study should be concise, clear, and to the point. Not every calculation and figure that has been generated, or can be generated, has any meaning or usefulness to demonstrate the value of the study. There is a lot of redundancy and irrelevant figures and tables included, which make no point.
The quality and style of presenting figures and results are very poor. Unnecessarily large figures can be avoided, for example, figures 13 and 14 can be simply shown as a matrix or table, or small figures. Some figures have poor resolution with visible pixelated numbers and text.
The style of presenting results needs to be improved drastically.
There are many new things, terms, and procedures that are described for the first time in the results section. This is very unusual and surprising. All the methods and procedures used should be well defined and described before we jump to results and analysis.
Clustering participants using PCA + K-Means (figure 6): It is very unclear why PCA and K-means are both used and for what purpose - from the hypothesis's point of view, this doesn't sit well under the flow of analysis that is required.
Paragraph at line 313: It is not clear how the description for each cluster is obtained (low Dark Triad, elevated), and more importantly, given the scores, why K-Means is required, why not simply group participants with low, moderate, and high with fixed thresholds.
Table 2: One of the columns says "Mean" and the heading says, average demographic and psych..cluster(), is that a typo?
Table 3, indicates the baseline - something that should be explained in the procedure, not in the result section.
Response: We sincerely thank the reviewer for drawing attention to the unintended introduction of new terminology and analytical procedures within the Results section. We fully agree that all methodological details should be presented beforehand to ensure clarity and a coherent narrative flow. In response to this concern, we have implemented the following revisions:
We have moved all methodological elements that were previously introduced in the Results section to the appropriate subsections of the Methods. This includes the descriptions of PCA, K-means clustering, data preprocessing, and the criteria used for cluster labeling. These revisions ensure that no new methodological information appears for the first time in the Results.
We have added and clarified the rationale for using the PCA + K-means approach. Specifically, PCA was applied to reduce dimensionality, minimize redundancy among psychological trait measures, and prevent multicollinearity. K-means clustering was then performed on the resulting component scores to identify naturally emerging psychological profiles without imposing arbitrary cutoffs. This analytical strategy directly supports Hypothesis 4, which proposes that distinct personality–stress configurations are associated with different psychophysiological response patterns.
Cluster descriptions have been clarified and more rigorously justified. Subjective labels such as “low Dark Triad” or “elevated” have been removed unless supported by objective criteria. Cluster labels are now defined based on z-scored PCA component weights rather than interpretive judgments. Additionally, we have included a brief explanation of how cluster centroids indicate relatively higher or lower scores along each principal component.
We have added a justification for selecting K-means rather than applying fixed thresholds. Fixed thresholds impose arbitrary cutoffs and do not account for the natural variability within the data, whereas K-means identifies data-driven groupings that more accurately reflect the structure of the sample. To support this choice, we conducted silhouette and inertia analyses, which indicated that a three-cluster (k = 3) solution provided a meaningful and robust representation of the data.
Comment: 3.4 Cardiovascular ....[]
Line 523: Understanding how ..threat. Please rewrite.
Unclarity of Set 1 to Set 6: The first time "Set" is mentioned in the manuscript in line 342, in the results section.
Correlation among psychological scores: Is there a reason to perform that, and can this be validated that the achieved correlation is as expected? As for predefined scores and metrics, there are established relationships. I miss seeing the value of this analysis.
Figures 7, 8, 9, and 10. Needs improved analysis. While computing correlation and plotting clusters doesn't inform anything.
Figure 12: Line plot is not appropriate for sets - categorical values.
Response: We have rephrased for clarity:
Rewritten for clarity: “Understanding how different cognitively and emotionally salient stimuli influence autonomic responses provides important insight into the physiological mechanisms underlying moral reasoning, self-presentation, and perceived identity threat. In this part of the study, heart rate (HR) and HRV-based stress scores were recorded before and after each question set to assess cardiovascular changes associated with specific emotional and cognitive demands”
Thank you for bringing this to our attention. We agree that introducing the term “Set” for the first time in the Results section could lead to confusion. To address this issue, we have added a clear explanation of the six question sets (Set 1–Set 6) in the Procedure subsection of the Methods, along with brief descriptions of the purpose of each set (e.g., neutral questions, personal-history questions, moral dilemmas, social-image questions, personality-relevant items, and deceptive responses).
We have also clarified the rationale behind correlating the psychological scores. The revised text now explains that these correlations were conducted to verify well-established relationships among the measured constructs, such as the positive associations among Dark Triad traits, their typical weak or negative associations with empathic responsiveness, and the higher emotional reactivity expected in individuals reporting elevated perceived stress.
This step served as a validity check to ensure that the psychological instruments operated as expected within our sample. Confirming these known associations provides confidence in the reliability of the measures and offers an appropriate context for interpreting the subsequent physiological and EEG patterns.
Following your suggestion, we have reorganized Figures 7, 8, 9, and 10 into two consolidated figures to improve clarity and reduce redundancy. We also revised and compacted the accompanying text to enhance readability and ensure a more coherent presentation of the findings. In addition, we clarified the rationale behind the analyses, explaining why autonomic responses were examined in relation to Dark Triad traits and how these patterns inform the interpretation of deceptive and neutral conditions. These updates strengthen the structure and interpretability of the results section.
Comment: Conclusion
The conclusion is relatively easy to read and follow.
Lines 881-882. The findings indicate..., that sentence should be followed by what that means. And needs improvement. It still needs to be improved.
Response: We have revised the text for greater clarity and added interpretative context to better explain how the results support the view of deception as a dynamic regulatory process.
“Deception elicits diverse physiological and neural responses shaped by individual personality and stress profiles. Autonomic and cortical data together reveal that lying is not purely a stress reaction but an adaptive process integrating emotional control and self-regulation.”
Reviewer 2 Report
Comments and Suggestions for Authors
This manuscript reports aninteresting investigation relatig factors in the Dark Triad to psychphysiological and neurological responses duirng deception. The authors are careful in their presentation ofthe methodology and the results are extensively reported. I take issue, howeverm, with the statement that the sample suice (n =30) is adeqaure from which to draw conclusions. this sample, comprising 2/3 of the sample female, does present an insufficient size from which to draw the conclusions as stated. this size of sample would not tasday past muster in any psychological journal. Added to this, the study used a repeated measures maninipulation, with counterbalancing of the stimuli across participants. While,counterbalancing of conditions may be asppropriate in some studies, with the manipulatuon of neutrasl and deceptive conditions there is a high probability that contamination of the effect of neutrality of condition after that of deception cannot be eliminated.
This study presents an interesting case, relating deception to the interaction of personality and stress. There needs, however, in my opinion, to be a replication of the methodology without counterbalancing of conditions and using a between conditions manipulation of deceit. An increasse in sample, with greater attention to the balance of geneder in the sdamples, would be a necessary addition.
The authoprs are to be congtratulated on their attention to the details of method and presentation of the results, but the sample, size and composition, does limit the conclusions which may be drawn. Also, while the authors do address limitations to the design of the study, there is no mention of the need to address the reactivity of participants, especially in a repeated measures design, to the manipulation of deception and ther contrast with neutrality, which may exacerbate or attenuate the impact of the manipulation. There does not appear to be any citation of methodological studies which address such reactivity effects. Any future study must examine the possible impact of ther ability of participants to see and interpret what it is the investigators are seeking to study and to find.
Author Response
Comment: This manuscript reports an interesting investigation related to factors in the Dark Triad to psychophysiological and neurological responses during deception. The authors are careful in their presentation of the methodology, and the results are extensively reported.
Response: We sincerely thank the reviewer for the positive feedback. Your comments have been highly valuable in strengthening the scientific relevance and overall contribution of our study. To further improve clarity and coherence, we have implemented additional revisions as outlined below:
Comment: I take issue, however, with the statement that the sample size (n =30) is adequate from which to draw conclusions. this sample, comprising 2/3 of the sample female, does present an insufficient size from which to draw the conclusions as stated. This size of sample would not be passed muster today in any psychological journal.
Response: We appreciate the reviewers’ comments on this important methodological aspect. We fully recognize that the sample size (n = 30) is modest and constrains the generalizability of our results. Accordingly, we have revised the manuscript to clearly indicate the limitations associated with the sample size and to provide explicit statements reflecting this. We have added the following:
"Given the modest sample size (n=30) and gender imbalance (70% female), this study should be considered an exploratory pilot investigation. The findings offer preliminary insights into psychophysiological patterns related to deception and emotional processing and should not be generalized beyond the characteristics of the current sample."
“Due to the limited sample size (n = 30), all analyses should be interpreted with caution, as the reduced statistical power increases the likelihood that subtle effects may not have been detected. As such, the present findings are best regarded as preliminary.”
“A major limitation of the present study is the modest sample size and the gender imbalance within the recruited cohort. While sample sizes of 20–40 participants are common in pilot psychophysiological and EEG research, they do not support definitive conclusions or broad generalization. As such, the cluster analyses and trait–physiology associations should be interpreted as exploratory patterns rather than robust effects. Replication in substantially larger and more demographically balanced samples is necessary to validate these trends and enhance the reliability of the findings.”
“Given the pilot nature of this work, the reported findings should be interpreted cautiously and validated through future studies using larger, more representative samples and higher-powered designs.”
Comment: Added to this, the study used repeated manipulation measures, with counterbalancing of the stimuli across participants. While counterbalancing of conditions may be appropriate in some studies, with the manipulation of neutral and deceptive conditions there is a high probability that contamination of the effect of neutrality of condition after that of deception cannot be eliminated.
Response: We sincerely thank you for this thoughtful and important observation. We fully agree that in paradigms involving both neutral and deception conditions, potential carryover or contamination effects represent a meaningful methodological concern. Although counterbalancing helps distribute order effects across participants, it cannot entirely rule out the possibility that exposure to a deception block may influence responses in a subsequent neutral block—for example, through heightened arousal, increased cognitive load, or strategic adjustments.
“Although conditions were counterbalanced across participants to mitigate order effects, this approach cannot fully eliminate potential contamination between the deception and neutral conditions. Specifically, completing a deception block first may heighten arousal, cognitive effort, or strategic monitoring, which could influence responses during a subsequent neutral block.”
“An additional limitation involves the potential contamination between the neutral and deception conditions. Although the presentation order was counterbalanced, exposure to a deception block may have influenced behavior or psychophysiological responses during a subsequent neutral block—potentially through sustained arousal, heightened vigilance, or increased cognitive control demands. While counterbalancing helps mitigate such carryover effects, it does not fully eliminate them. Future studies should consider using between-subjects designs, longer washout intervals, or independent condition presentations to more effectively prevent contamination.”
Comment: This study presents an interesting case, relating deception to the interaction of personality and stress. There needs, however, in my opinion, to be a replication of the methodology without counterbalancing of conditions and using a between conditions manipulation of deceit. An increase in samples, with greater attention to the balance of gender in the samples, would be a necessary addition.
Response: We sincerely appreciate your constructive feedback and fully concur with these recommendations. This study was conceived as an initial exploration of psychophysiological and neural responses to deception in relation to personality traits. Although counterbalancing helped reduce order effects, we acknowledge that a between-subjects design comparing deception and neutral conditions would more effectively prevent potential contamination. In response to your comment, we have added the following clarification to the manuscript:
"A recommendation for future research is to replicate these findings using a between-subjects design for the deception manipulation, rather than a within-subjects counterbalanced approach. This would eliminate potential carryover effects between neutral and deceptive conditions, allowing for a clearer interpretation of condition-specific psychophysiological responses. Additionally, future studies should aim to recruit larger, more gender-balanced samples. A more demographically diverse cohort would increase statistical power, support more robust analyses of personality–stress interactions and improve the generalizability of the results."
Comment: The authors are to be congratulated on their attention to the details of method and presentation of the results, but the sample, size and composition, does limit the conclusions which may be drawn. Also, while the authors do address limitations to the design of the study, there is no mention of the need to address the reactivity of participants, especially in a repeated measures design, to the manipulation of deception and the contrast with neutrality, which may exacerbate or attenuate the impact of the manipulation.
Response: Thank you very much for your positive assessment of the manuscript. To address your concerns, we have added text to the Discussion section highlighting the sample constraints, the possibility of participant reactivity due to repeated manipulations, and the need for future studies employing between-subjects designs to fully eliminate reactivity-related confounds. We have added the following:
“Beyond sample-related limitations, repeated exposure to both deceptive and neutral conditions may have affected participant engagement, stress responses, or expectations. Such reactivity effects could either reduce the contrast between conditions or, alternatively, increase sensitivity to deceptive trials. Recognizing these dynamics is important for interpreting the current findings and underscores the need for future studies to adopt designs that minimize cross-condition contamination.”
Comment: There does not appear to be any citation of methodological studies which address such reactivity effects.
Response: We have added the following references:
Hertwig, R., & Ortmann, A. (2008). Deception in experiments: Revisiting the arguments in its defense. Ethics & Behavior, 18(1), 59–92. https://doi.org/10.1080/10508420701712990
Meijer, E. H., Verschuere, B., Gamer, M., Merckelbach, H., & Ben-Shakhar, G. (2016) Deception detection with behavioral, autonomic, and neural measures: Conceptual and methodological considerations that warrant modesty. Psychophysiology. 53(5), 593-604. https://doi.org/10.1111/psyp.12609
Lee, S., Niu, R., Zhu, L., Kayser, A. S., & Hsu, M. (2024) Distinguishing deception from its confounds by improving the validity of fMRI-based neural prediction. Proceedings of the National Academy of Sciences of the United States of America, 121(50), e2412881121. https://doi.org/10.1073/pnas.2412881121
Comment: Any future study must examine the possible impact of the ability of participants to see and interpret what it is the investigators are seeking to study and to find.
Response: In response to your comment, we have expanded the Discussion and Limitations sections as follows:
“Another important methodological consideration involves the possibility that participants may infer the study’s purpose and adjust their behavior accordingly. Research on demand characteristics and participant awareness of experimental aims (Orne, 1962) indicates that such awareness can influence both behavioral and physiological responses. In deception paradigms, participants may become increasingly sensitive to what they believe is being evaluated, potentially amplifying, suppressing, or otherwise distorting the intended manipulation.
To address this, future studies should incorporate explicit assessments of participant awareness—such as post-experimental questionnaires, funneled debriefings (Bargh & Chartrand, 2000), or structured manipulation checks—to evaluate the extent to which participants detect the study’s purpose. Systematically examining these reactivity factors will help distinguish genuine deception-related effects from those shaped by participants’ interpretations of the research objectives.”
Round 2
Reviewer 2 Report
Comments and Suggestions for Authors
The authors have addressed all of my concerns and have outlined in their presentation ways in which the procedures may be improved in future studies and have also addressed them in such a way that any reader will be aware of the restrictions that must be apploed in drawing defintiive conclusions from the data.
In light of these quite considerable additions to the manuscript, I would recommend that the paper proceed to publication.
Author Response
Comment: The revised version shows a clear improvement in terms of theoretical framing, methodological detail, and discussion of the pilot nature and limitations of the study. From a scientific standpoint, the manuscript is now suitable for publication in the Special Issue. However, I recommend a final round of minor revisions focused on clarity and style:
Response: Thank you for your valuable and constructive feedback. We have carefully considered and addressed each of your comments. Below, we provide a detailed summary of the revisions made in response to the reviewers’ remarks. In the manuscript, all major modifications are highlighted in blue for easier identification.
Comment: Please remove redundant paragraphs in the Methods section (e.g., duplicated descriptions of the participant sample, Garmin-based HRV/stress measurement, and EEG recording) and correct the inconsistent section numbering (e.g., “2.6. 2.5.”, “2.8. 2.7.”).
Response: The Methods section has been revised to eliminate repeated descriptions of the participants, Garmin HRV measurements, and EEG recording procedures. All inconsistencies in section numbering (e.g., “2.6. 2.5.”) have been corrected.
Comment: Harmonize verb tenses in the Methods (past tense for procedures already conducted) and streamline the Abstract to avoid repeated sentences describing the same results.
Response: The Methods section has been revised to consistently use the past tense for completed procedures, and the Abstract has been updated to remove repetitive phrasing and improve overall clarity.
Comment: Consider a light language editing throughout the manuscript to improve readability and to standardize the reporting of statistics (e.g., p-values).
Response: A general language revision was carried out throughout the manuscript to improve readability, and the statistical reporting—including the formatting of p-values—has been standardized.
Comment: In the Discussion, please explicitly stress that the cluster analysis is exploratory and not intended to provide robust typologies given the small sample size (N=30).
Response: We have revised the Discussion to clearly state that the cluster analysis is exploratory given the small sample size (N = 30) and should not be interpreted as producing stable typologies.
"Although the cluster analysis suggested the presence of two participant groups with differing psychophysiological patterns, this analysis is exploratory in nature and should be interpreted with caution. Given the small sample size (N = 30), the resulting clusters do not represent stable or generalizable typologies but instead provide preliminary insights to guide future research."